# Development and Characterization of Cationic Nanostructured Lipid Carriers as Drug Delivery Systems for miRNA-27a

**DOI:** 10.3390/ph16071007

**Published:** 2023-07-14

**Authors:** Amina Tucak-Smajić, Ivana Ruseska, Ilse Letofsky-Papst, Edina Vranić, Andreas Zimmer

**Affiliations:** 1Department of Pharmaceutical Technology, Faculty of Pharmacy, University of Sarajevo, Zmaja od Bosne 8, 71000 Sarajevo, Bosnia and Herzegovina; amina.tucak@ffsa.unsa.ba (A.T.-S.); edina.vranic@ffsa.unsa.ba (E.V.); 2Department of Pharmaceutical Technology and Biopharmacy, Institute of Pharmaceutical Sciences, University of Graz, Universitätsplatz 1, 8010 Graz, Austria; ivana.ruseska@uni-graz.at; 3Institute of Electron Microscopy and Nanoanalysis, Center for Electron Microscopy, Graz University of Technology, NAWI Graz, Steyrergasse 17, 8010 Graz, Austria; ilse.papst@tugraz.at

**Keywords:** drug delivery system, miRNA-27a, cationic lipid nanoparticle, anti-adipogenic effect, 3T3-L1 cell, cellular uptake

## Abstract

Although miRNA-27a has been identified as a promising candidate for miRNA mimic therapy of obesity, its application is limited due to enzymatic degradation and low membrane permeation. To overcome these problems, we developed cationic nanostructured lipid carriers (cNLCs) using high-pressure homogenization and used them as non-viral carriers for the anti-adipogenic miRNA-27a. Cargo-free octadecylamine-containing NLCs and miRNA/cNLC complexes were characterized regarding particle size, size distributions, zeta potential, pH values, particle topography and morphology, and entrapment efficacy. Furthermore, the cytotoxicity and cellular uptake of the miRNA/cNLC complex in the 3T3-L1 cell line were investigated. The investigation of the biological effect of miRNA-27a on adipocyte development and an estimation of the accumulated Oil-Red-O (ORO) dye in lipid droplets in mature adipocytes were assessed with light microscopy and absorbance measurements. The obtained data show that cNLCs represent a suitable DDS for miRNAs, as miRNA/cNLC particles are rapidly formed through non-covalent complexation due to electrostatic interactions between both components. The miRNA-27a/cNLC complex induced an anti-adipogenic effect on miRNA-27a by reducing lipid droplet accumulation in mature adipocytes, indicating that this approach might be used as a new therapeutic strategy for miRNA mimic replacement therapies in the prevention or treatment of obesity and obesity-related disorders.

## 1. Introduction

The rising prevalence of overweight and obesity has become a pressing global health issue often connected to chronic conditions, such as diabetes mellitus, dyslipidemia, cardiovascular diseases, cancers, and non-alcoholic fatty liver disease [1,2]. The current approach to treating or preventing obesity is based on stopping lipid accumulation or/and increasing energy expenditure [3]. Nowadays, nucleic acid-based therapy has emerged as a promising approach for a majority of inherited or acquired diseases and conditions, including obesity, by transferring coding or non-coding gene sequences to produce proteins required for the establishment or maintaining metabolic homeostasis [4,5].

MicroRNAs (miRNAs) represent an important class of small, double-stranded non-coding RNAs, discovered in 1993, composed of approximately 21–23 nucleotides. They regulate gene expression at the post-transcriptional level through the repression of translation or degradation [6,7,8] in several cell types, such as adipocytes, β-cells, and muscle cells [9]. miRNAs play a significant role in metabolic diseases such as obesity and diabetes, regulating adipose functions linked with obesity [3,9]. In brief, adipocyte differentiation is promoted by miR-14, while insulin secretion in pancreatic islet cells is regulated by miR-375 [10]. It was found that miRNA-27a inhibits adipogenic differentiation by downregulating adipogenic marker genes such as PPARγ (peroxisome proliferator-activated receptor γ). This adipogenic factor therefore cannot trigger endogenous CCAAT-enhancer-binding protein α (C/EBPα) transcription, which results in the inhibition of adipocyte differentiation [11]. This study has drawn attention to the fact that miRNA-27a might be employed in miRNA mimic replacement therapy to prevent or treat obesity [2,11].

Effective cellular internalization and intracellular release are especially important for miRNA-based therapies, which need to be delivered to the cytosol to fulfil their bioactivity [12]. Despite the great potential shown by nucleic acid-based treatments, the application of naked miRNAs, which are hydrophilic molecules, faces numerous obstacles, such as the low permeability of the cell membrane, rapid enzymatic degradation in the bloodstream, and tissue non-specificity [13,14,15]. Hence the need for the design of a drug delivery system (DDS) to overcome the previously mentioned obstacles and deliver miRNAs into cells, which include viral and non-viral carriers [16].

Viral carriers were developed for gene delivery due to their natural ability to infect cells, but they meet safety problems due to the possibility of inducing immune responses that can eliminate transgene expression [17]. As viral ones have shown problems associated with low loading capacity, lack of quality control, and limitations linked to clinical application and large-scale manufacturing, non-viral carriers, such as lipid nanoparticles, have emerged as potential carriers for miRNA [16,18].

The development of lipid nanoparticles began in the early 1990s [19], and over time, they have provided an alternative to organic nanoparticles and prior lipid-based formulations like liposomes [20]. As a matrix composed of solid lipids only, these particles were different from previously established nanoemulsions or liposomes, and they were called solid lipid nanoparticles (SLNs) [19].

Later, the second generation of lipid particles, known as nanostructured lipid carriers (NLC), was introduced as a modified form of SLNs [19,21]. NLCs are derived from SLNs when one part of the lipid phase is replaced by a liquid lipid. While solid lipid(s) make up 0.1–30% (*w*/*w*) of SLNs, NLCs can have a total lipid content of up to 95% (*w*/*w*) [22]. Unlike SLNs, NLCs have an unstructured lipid matrix composed of a binary mixture of two spatially different solid lipid matrices (solid lipid and liquid lipid) with different melting temperatures, in a ratio of 70:30 to 99.9:0.1 [23]. The lipid matrix of NLCs remains solid without forming an ordered crystal lattice, which prevents the expulsion of small lipophilic drug molecules from the lipid matrix [24].

As NLCs are composed of a lipid matrix stabilized by surfactants, cationic NLCs (cNLCs) also contain at least one cationic lipid, such as octadecylamine (OA), 1,2-dioleoyl-3trimethylammonium-propane (DOTAP), 3β-[N-(N’, N’-dimethylaminoethane)-carbamoyl]cholesterol (DC-Chol), etc. [25]. The incorporation of a cationic lipid into the lipid matrix plays a critical role in the self-assembly of miRNA/cNLC complexes, termed NLCplexes, through the electrostatic interaction between negatively charged miRNA molecules and positively charged lipid particles [26] and the further cellular uptake of the positively charged complexes by the interaction with the negatively charged cellular membranes [27].

Cationic lipid nanoparticles had been employed for the delivery of mRNA molecules, such as in vaccines for COVID-19 (for example, Pfizer-BioNTech (BNT162b2) and Moderna (mRNA-1273) vaccines). However, these DDS are produced by mixing lipid stock solutions in ethanol with an aqueous solution of mRNAs in microfluidic mixers [28], which results in mRNA encapsulation inside the lipid nanoparticles [29].

In our case, we aimed to produce cargo-free cationic NLCs that will be used as carriers for miRNA molecules in a later step. Given that it is estimated that <2% of nucleic acids administered in lipid nanoparticles escape from the endosomes to reach the cytosol [30], in our DDS, miRNA is adsorbed at the particle surface through electrostatic interactions. This approach should enhance the interactions with cellular membranes, and the endosomal escape of miRNA, as the degradation of NLCs is not necessary to release the cargo because these molecules can be released by dissociation from the particles’ surface.

So far, for miRNA-27a, cell-penetrating peptides have shown excellent potential for developing DDS through spontaneously induced self-assembly complexations [31], but particles have shown the agglomeration tendency through peptide–peptide interactions, due to their hydrophobic amino acid sequence. Previous research has shown that lipid-based nanoparticles are stable and biocompatible, and that their uptake into the body can overcome physicochemical and biological barriers, ultimately leading to high transfection efficiency [16,32,33]. In several studies, cationic NLCs were used as DDS for miRNA.

For example, Chen et al. reported that cNLCs were successfully applied to deliver miR-34a in experimental lung metastases of mouse B16F10 melanoma. After treatment, tumor cell migration was significantly inhibited in vivo, indicating that cationic NLCs have in vivo potential for miRNA delivery [34]. Also, cationic NLCs have been used to deliver miR-107 in the treatment of head and neck squamous cell carcinoma in vitro and in vivo [35]. However, cationic NLCs were not reported to be used as carriers for miRNA-27a.

Due to their advantages as non-viral carriers, the main aim of this study was to design and optimize OA-containing NLCs as DDS for miRNA-27a. The incorporation of OA in lipid nanoparticles is commonly described, since this cationic lipid possesses a positively charged primary ammonium group within a wide pH range [36]. Nucleic acids should spontaneously interact with OA molecules and result in the protection of nucleic acids associated with lipid nanocarriers [5,37]. Cationic lipids are responsible for the interaction with negatively charged miRNAs, and they affect the nanoparticle’s properties, such as the surface charge/pKa (pKa of OA is 10.6) [5,38]. In general, the cationic lipid should be able to adjust its charge after systemic delivery depending on the pH of the environment, such as the bloodstream (pH 7.4) and endocytic organelles (pH 5.5–6.5), to enable the release of miRNAs into the cytosol, which is often a rate-limiting step for the successful delivery of miRNA in vivo [38].

Herein, we demonstrated that cNLCs are suitable DDS for anti-adipogenic miRNA-27a. The cargo-free cNLCs and miRNA/cNLC complex were characterized in terms of particle size, polydispersity index (PdI), zeta potential (ZP), entrapment efficiency (EE), storage stability, particle topography and morphology, and cytotoxicity. In vitro transfection experiments of the miRNA/cNLC complex are performed using 3T3-L1 cells to investigate the degree of lipid droplet formation during adipocyte differentiation and to evaluate the biological effect of miRNA-27a in mature adipocytes. Furthermore, cellular uptake studies are performed to investigate the efficiency of miRNA/cNLC complexes in interacting and delivering the miRNA into cells.

## 2. Results

### 2.1. Development of Nanostructured Lipid Carriers by High-Pressure Homogenization

The NLC formulations that contain increasing concentrations of OA (0–0.5%, *w*/*w*) were produced by a high-pressure homogenization process and characterized in terms of particle size, PDI, ZP, and pH values at the production date (Table 1).

The HPH process yielded NLCs with a mean particle size from ~105 nm to 116 nm, with a relatively narrow size distribution, as PdI values varied from 0.180 to 0.218. Compared to the control NLC formulation (bNLC), the gradual addition of OA in the NLC matrix led to a rise in particle sizes of the cNLC formulation, which was more pronounced at OA concentrations above 0.15% (*w*/*w*) (*p* < 0.001). The considerable increase in size distribution (*p* < 0.05), given as PdI values, was noticed in all formulations except for cNLC–1. On the other hand, similar values for d_(0.5)_, d_(0.9),_ and d_(0.99)_ of the produced cNLC formulations were obtained by the LD measurements, regardless of OA concentration. Furthermore, no signs of bigger particles or agglomerates were found in NLC formulations (Table 1).

The zeta potential values of the produced NLC formulations are presented in Table 1. As expected, the bNLC had negative ZP values (−18.2 ± 1.2 mV), due to the presence of non-ionic surfactants. However, with the gradual addition of the cationic lipid to the NLC formulations, the ZP values changed to highly positive ZP values of 46.8 ± 2.1 mV (cNLC–6), indicating the successful production of cationic NLCs. Except for cNLC–1, all cNLC formulations had ZP values > 30 mV, which is required not only for appropriate miRNA binding, but also for promoting the efficient stabilization of the NLCs. Furthermore, a minimal increase in ZP values was noticed in all cNLC formulations containing OA in concentrations higher than 0.15% (*w*/*w*). These findings are similar to those already published, particularly in formulations containing solid lipids, cationic lipids, and surfactants, due to the possible saturation of the particle interface by OA [39]. On the other hand, the pH of cNLCs formulations was raised to approximately 10 (Table 1) by adding OA. This result may be explained by the presence of the primary ammonium group in the OA structure [39]. Therefore, the pH of cNLC was adjusted to ~7 to produce theoretically fully ionized OA molecules at the interface of NLC particles. This practice is followed in many studies that use OA as a cationic lipid in their formulations [5]. The stability of cargo-free cNLC formulations without pH adjustment was also assessed; however, the stability of cNLC formulations was significantly lower due to the rapid decline in ZP values. This step is not necessary when using ternary or quaternary cationic lipids, such as DDAB or CTAB.

### 2.2. Cytotoxicity of Nanostructured Lipid Carriers

Given that cationic lipids may have cytotoxic effects due to their interference with cell or subcellular compartment membrane function and integrity [40], MTS and LDH tests were performed to evaluate the cell viability and toxicity of NLC formulations containing OA (0–0.5%, *w*/*w*) at dilutions from 1:10 to 1:250 with serum-free lgDMEM.

As shown in Figure 1A,B, dose-depended cytotoxic profiles of NLC formulations were reached at OA concentrations above 0.15% (*w*/*w*) (*p* < 0.001), whereas cNLC–1, cNLC–2, and cNLC–3 had similar profiles as the bNLC formulation. Therefore, despite their excellent physicochemical properties, cNLC–4, cNLC–5, and cNLC–6 formulations were excluded from further studies due to the high cytotoxicity effect seen in 3T3-L1 cells.

### 2.3. Stability Studies of Nanostructured Lipid Carriers

The physicochemical properties of selected NLC formulations were evaluated after storage in transparent glass vials at three different temperature conditions—5 ± 3 °C, 25 ± 2 °C (60 ± 5% RH), and 40 ± 2 °C (75 ± 5% RH)—for a defined storage period of 9 months. At predefined times, the formulations were characterized in terms of particle size, PdI, ZP, pH values, and visual appearance (Figure 2 and Appendix A).

As shown in Figure 2A, all NLC formulations maintained a relatively similar visual aspect, particle size, and PdI over nine months of storage in the fridge, while a sharp decrease in ZP of cNLC–2 was observed (*p* < 0.05) after 270 days (Figure 2B). During storage at 25 ± 2 °C, all formulations were stable for at least six months (Figure 2C). The particle size of cNLC–2 increased after nine months due to the low ZP values of ~8.6 mV, as presented in Figure 2D. The DLS data reveal that only the particle size of the cNLC–3 formulation was maintained unchanged for 270 days, regardless of the storage temperature. In general, as the storage temperature increased, the ZP values of cNLCs decreased gradually over nine months (Figure 2B,D,F). This decrease in the ZP values can be attributed to the coverage of the positively charged NLC interface with the negatively charged free fatty acids and interaction with OA molecules. As the ZP of bNLC increases, the coverage of anionic NLC particles with negatively charged free fatty acids may become more pronounced. This phenomenon can have the opposite effect, as both the fatty acids and the particle interface carry negative charges.

Simultaneously with the drop in ZP values, the pH of the formulations was reduced, most likely due to the formation of free fatty acids from the oxidation and hydrolysis of MCT. The higher the concentration of OA in the NLC formulation, the greater the decrease in pH over time. As the cNLC–3 formulation has shown an excellent physical stability profile, this formulation was used for further investigations and the preparation of miRNA/cNLC complexes.

### 2.4. Preparation of miRNA/cNLC Complexes

The selected cNLC–3 formulation was used for the preparation of miRNA/cNLC complexes, termed NLCplexes. Therefore, the cNLC formulation was freshly prepared and mixed with a working solution of miRNA-27a in different mass ratios between miRNA and OA from the cNLC formulation. The obtained complexes contained higher amounts of miRNA-27a (10:1, 5:1, and 2.5:1, *w*/*w*), equal amounts of miRNA-27a and cationic NLC (1:1, *w*/*w*), and higher amounts of cationic NLC formulation (1:2.5; 1:5, and 1:10, *w*/*w*). To be suitable for DLS and ELS measurements, the miRNA concentration in all sample conditions was 100 nM. Figure 3A depicts the particle size, expressed as z-average and size distribution (PdI), of the formed NLCplexes, whereas the ZP values and entrapment efficacy of miRNA are shown in Figure 3B.

DLS studies have revealed that the particle size of the obtained NLCplexes increased approximately twice (above 200 nm) in mass ratios from 10:1 to 2.5:1, compared to the cargo-free cNLC–3 formulation at the day of production (~113 nm). This increase in particle size could be related to the insufficient complexation between miRNA-27a and cNLC. However, at a mass ratio of 1:1, the particle size and PdI increased dramatically to 571.7 ± 41 nm due to the low ZP values of 8.30 ± 0.15 mV, which resulted in particle instability. As the mass ratio rose from 1:2.5 to 1:10, the particle size sharply fell to 120–130 nm, similar to the particle size of the cNLC–3 formulation before complexation with miRNA.

The development of NLCplexes in different mass ratios resulted in alterations in ZP values and the inversion of negative (−25.3 ± 1.1 mV) to highly positive (42.9 ± 0.6 mV) values (Figure 3B). Interestingly, a saturation process was observed at higher mass ratios (beyond 1:2.5), where the ZP reached a plateau with no discernible variations in ZP, most likely due to an excess of cNLC particles. The positive ZP values of the complex indicate the complete neutralization of miRNA, which is not only required for the interaction with the cell membrane and the initiation of endocytosis, but also for the promotion of complex stability by electrostatic repulsion.

The entrapment efficacy of miRNA-27a to cNLC–3 was estimated with the RP-HPLC method, which determines the concentration of unbound miRNA after the ultracentrifugation of NLCplexes. As depicted in Figure 3B, gradually adding miRNA to cNLC particles improved the entrapment efficacy from 2.27% (10:1, *w*/*w*) to 85.65% (1:5, *w*/*w*) and 97.87% (1:10, *w*/*w*). A gel electrophoresis assay verified that total complex formation could be achieved above the miRNA/OA mass ratio of 1:1. This result confirmed the binding ability of miRNA-27a to cNLCs (Figure 4). As expected, when the mass ratio was above 1:1, no free miRNA-27a could be seen in the gel band, indicating that complexes can retain added miRNA at these mass ratios.

### 2.5. Complexation with Different Types of miRNAs

Additional DLS and ELS tests were conducted to investigate the complexation behavior of different types of miRNAs used in in vitro experiments with the cNLC–3 formulation. The complexes with miRNA-NTC and miRNA-FluoNTC were prepared with the cNLC–3 formulation in three different mass ratios of 5:1, 1:2.5, and 1:5, containing 100 nM of miRNA (Figure 5).

All types of miRNAs (miRNA-27a, NTC, and FluoNTC) are double-stranded with molecular weights ranging from 13,454 g/mol to 15,891 g/mol, and chain lengths from 19 to 24 nucleotides. The particle size measurements of NLCplexes obtained with the three tested nucleic acids (Figure 5A) revealed similar physicochemical properties of the obtained complexes, regardless of the used type of miRNA (*p* > 0.05). In brief, the particle size of 200–250 nm was obtained in NLCplexes at a mass ratio 5:1, whereas a particle size of 130–160 nm was reached in NLCplexes containing higher amounts of cNLC particles (mass ratios 1:2.5 and 1:5). Additionally, the ZP values were around −20 mV at a mass ratio of 5:1 due to the excess of miRNA molecules, and were highly positive (above 40 mV) in mass ratios 1:2.5 and 1:5, as presented in Figure 5B.

### 2.6. Morphology and Topographic Studies

The morphological and topographic analyses were carried out by cryo-TEM and AFM methods on the selected cNLC–3 formulation and the complex of this formulation with miRNA-27a (Figure 6).

### 2.7. Stability Studies of miRNA/cNLC Complex

Figure 7 shows a short-term stability study that tracked changes in the physicochemical properties of the miRNA-27a/cNLC complex at mass ratios of 5:1 and 1:5 during 30 days of storage in the fridge (5 ± 3 °C). After predetermined periods, the NLCplexes were characterized in terms of particle size, PdI, and ZP by DLS and ELS methods.

The DLS data show the fluctuations in particle size over time for the NLCplex in an miRNA/OA mass ratio 5:1 (Figure 7A). The particle size decreased from 261.7 ± 24.9 nm, obtained immediately after the preparation of the complex, to ~170 nm, which was maintained for 2 h. Then, particle size increased to ~212 nm after 3 h, and this profile was maintained for one day. However, the particle size continued to rise during the next ten days to around 290 nm, and unexpectedly started to decrease after 20 days to 202 nm, followed by a decrease in the next ten days to a final particle size of 183.7 ± 5.6 nm. Throughout 30 days, the PdI values were 0.19–0.35, with no significant changes over one month of storage (*p* > 0.05).

On the other hand, the NLCplex at a mass ratio 1:5 maintained the particle size in the range of 155–173 nm over ten days (*p* > 0.05), while gradual increases to ~183 nm and 210 nm were observed after 20 days (*p* < 0.05) and 30 days (*p* < 0.001), respectively. During this period, the PdI values varied from 0.16 to 0.24, but their differences were not considered significant (*p* > 0.05).

The ELS data reveal that the NLCplex in the mass ratio of 5:1 kept a negative surface charge from −20 mV to −24 mV for ten days, after which the ZP values decreased to −7.4 ± 0.6 mV (Figure 7B). The NLCplex at the mass ratio of 1:5 was capable of maintaining highly positive ZP values for around three days. A decreasing trend in surface charge was observed over time, and after 30 days, the ZP values decreased twofold compared to the freshly produced NLCplex (18.5 ± 1.2 mV).

It can be concluded that even though NLC carriers (cNLC–3 formulation) have shown excellent physical stability for nine months, the complex with miRNA can maintain its physicochemical properties for just a couple of days. The instability of the complexes can be attributed to several factors. First, the temperature may play a role, as miRNA solution should be stored at −80 °C, and in this condition, miRNAs are stable for at least 12 months. Therefore, the stability of free miRNAs is not assessed. Even though cNLCs are stable in the fridge, this temperature is probably unsuitable for the long-term stability of complexes. Additionally, despite using RNase-free water during complex preparation, the repeated measurements conducted over 30 consecutive days could potentially lead to miRNA degradation. Factors such as the use of pipette tips, zeta cuvettes, and other handling procedures may contribute to this degradation. It is important to note that the cNLC formulation was prepared using Milli-Q water, which may also affect miRNA stability. Therefore, our general recommendation is to prepare the complex ex tempore, to ensure that no significant changes in its physicochemical properties occur.

### 2.8. Cytotoxicity Study

The MTS and LDH assays were performed to evaluate the cytotoxic profile of the free cNLC formulation, miRNA-27a, and miRNA-NTC, as well as the formed miRNA-27a/cNLC complexes in mass ratios of 1:2.5 and 1:5 (*w*/*w*). All samples were diluted with lgDMEM to contain a final miRNA concentration from 25 to 100 nM. The obtained data are expressed as a percentage of cell viability (Figure 8A) and cell cytotoxicity (Figure 8B) compared to control cells.

The results show that the free miRNAs and NLCplexes possess low cytotoxic profiles. On the other hand, the cNLC–3 formulation in a concentration equivalent to those in NLCplexes showed a dose-dependent cytotoxicity effect in 3T3-L1 cells (Figure 8A). Nonetheless, the cell viability of all tested samples was still above or close to 70%. On the other hand, the LDH data show that, under all sample conditions, cell toxicity was below 20% (Figure 8B).

### 2.9. In Vitro Transfection and Differentiation of 3T3-L1 Cells

The miRNA/cNLC complexes were designed as DDS to deliver miRNA extracellularly into 3T3-L1 preadipocytes. Two types of NLCplexes containing 25–100 nM of either miRNA-27a or NTC in miRNA/OA mass ratios 1:2.5 and 1:5 were used to transfect cells to evaluate the anti-adipogenic effects of miRNA in cells.

The effects of these NLCplexes on the degree of lipid formation in mature adipocytes were assessed after ORO staining and the evaluation of staining intensities of the accumulated lipophylic ORO dye in lipid droplets with light microscopy (Figure 9) and absorbance measurements (Figure 10).

Furthermore, free miRNA-27a, NTC, and cargo-free cNLC formulations were applied to cells. Two types of cells were used as controls: the first control represents the differentiated cells, which received all hormonal substances to differentiate into mature adipocytes, whereas the other control represents cells that received only the proliferation medium and, therefore, did not differentiate into mature adipocytes.

The percentages of reduction in lipid accumulation in cells as compared to differentiated cells, which are considered as showing 100% of the accumulation of lipid droplets, are given in Table 2.

Figure 9 depicts the control samples, consisting of differentiated cells, cargo-free cNLC, free miRNA-27a, and NTC, and the experimental group consisting of miRNA(27a/NTC)/cNLC complexes. It can be noted that cells that received only induction and differentiation medium exhibit an intensive red coloring, which was also observed in cells transfected with free miRNA-NTC and miRNA-27a, regardless of the applied miRNA concentration. Compared to cells only, a concentration-dependent decrease in the ORO-staining intensity was noticed in cells transfected with free cNLC and NLCplexes. However, in both mass ratios, a more intense red coloring was observed in complexes that contain NTC, compared to complexes with miRNA-27a, which can be associated with the anti-adipogenic effect of miRNA-27a. The measured absorbance of ORO dye that accumulated in lipid droplets in differentiated cells was 0.167 ± 0.016, compared to the undifferentiated cells, which had much lower absorbance (0.065 ± 0.006), as shown in Figure 10A.

Cells transfected with free miRNA (27a or NTC) for all tested nucleic acid concentrations did not show a significant effect on the reduction in lipid droplet formation (*p* > 0.05) compared to cells only (Figure 10A). This result is consistent with the literature because it is known that, due to the negatively charged surface of miRNA, they repel negatively charged cell membranes electrostatically, resulting in a limited absorption of naked miRNA [41]. These results verify our hypothesis that miRNAs need a DDS as carriers to cross cell membranes.

On the other hand, a rise in the applied concentration of cargo-free cNLC formulation caused a gradual decrease in absorbance from 0.146 ± 0.021 to 0.104 ± 0.019. This result means that cNLC itself possibly affects adipocyte differentiation, probably because of the slightly cytotoxic effect of cNLC, and caused stress to cells due to the interactions of cationic nanoparticles with negatively charged membranes [5].

The NLCplexes in mass ratios 1:2.5 and 1:5 showed a dose-depended decrease in ORO accumulation and absorbance reduction. This effect was noticed even at the lowest miRNA-27a concentration (25 nM), where a significant decrease in absorbance to 0.135 ± 0.011 was measured compared to cells only (*p* < 0.001). With higher concentrations, this effect is more pronounced in miRNA-27a-containing complexes (1:2.5), where absorbance dropped to 0.126 ± 0.10 (*p* < 0.001) and 0.095 ± 0.021 (*p* < 0.0001) when the complexes contained 75 nM and 100 nM of miRNA-27a, respectively (Figure 10B).

Even though in NTC-containing complexes, a steady decrease in absorbance values is noticed, the effect of a reduction in lipid droplet formation in cells is more efficient in cells transfected with miRNA-27a/cNLC complexes (*p* < 0.05) containing the same concentration of miRNA. As miRNA-27a/cNLC complexes caused a reduction in lipid accumulation in cells for 20.20 ± 1.03% (75 nM) and 42.87 ± 2.11% (100 nM), this experiment supports our hypothesis that cNLC can be used as appropriate DDS for the successful delivery of miRNA-27a, which is confirmed via the anti-adipogenic effect that was caused during adipocyte differentiation. It is important to note that similar observations were reported in a study involving complexes with cationic peptides and miRNAs [42]. In this study, the observed effects seen with NTC were attributed to the influence of the carrier system on cellular signaling, as well as mechanical stress caused by the nanoparticles on the top of the cell membrane. However, these effects are unrelated to toxicity (unpublished data).

At an miRNA/cNLC mass ratio of 1:5, the effect of reduced lipid accumulation is even more noticeable. These results can be explained due to the slight impact of DDS itself on the process of adipocyte differentiation, as the concentration of miRNA was the same as in mass ratio 1:2.5. As shown in Figure 10C, the incubation of cells with miRNA-27a/cNLC complexes decreased the absorbance from 0.141 ± 0.03 (25 nM) to 0.085 ± 0.010 (100 nM), which led to a reduction in lipid formation for 24.37 ± 1.03% (75 nM) and 49.40 ± 0.97% (100 nM), as shown in Table 2.

The absorbance measurements should always be accompanied by light microscopy images. Due to the multiple pipetting steps (as can be seen from the protocol), some spots of the cell well can be damaged, which can influence the absorbance measurement results. As the measurement mode is a matrix (25 × 25), average absorbance is taken after measuring 25 spots in one well. Therefore, we took pictures of all wells under the light microscope to be sure that the lipid droplet reduction is visible.

### 2.10. Cellular Uptake Studies

The cellular uptake of FluoNTC/cNLC complexes in 3T3-L1 cells was evaluated at different time points after transfection (4 h and 24 h). Figure 11A depicts the influence of concentration on the uptake. For the FluoNTC/cNLC mass ratio 1:2.5, we can observe an increase in the taken-up fraction with time. At the first time point (4 h), there is a striking contrast in uptake at different concentrations. The uptake of 100 nM complexes is higher compared to the uptake of 200 nM complexes, with a difference of about 17%. This difference decreases with time—as we can observe, the taken-up fraction for both concentrations increases after 24 h, and it reaches a maximum of 41.4% and 33.5% for 100 nM and 200 nM, respectively.

For complexes containing a higher amount of cationic lipid (1:5), the situation is slightly different. The internalized fraction after 4 h seems to be around 10% for both concentrations, and it increases to 34.6% for 100 nM and 21.4% for 200 nM complexes 24 h after transfection. Similar to 1:2.5 complexes, in this case, we can also observe that the concentration of 100 nM seems to be more favorable for uptake.

Figure 11B demonstrates how the fraction of positively charged lipids in the complexes influences the uptake. Surprisingly enough, it seems that the mass ratio of 1:2.5 is preferred for overall cellular uptake. At a concentration of 100 nM, after 4 h, the taken-up fraction of 1:2.5 complexes is around 27%, whereas the fraction of 1:5 complexes is only 10%. With time, the uptake of both complexes increases, with a maximum of 41.4% and 34.6% for 1:2.5 and 1:5 complexes, respectively. The opposite occurs at a concentration of 200 nM. In this case, in the early stages of incubation, we can observe an uptake of 9.9% for 1:2.5 and 12.8% for 1:5 complexes. Nevertheless, the internalization of 1:2.5 complexes increases in time, to a maximum of 33.5%, while 1:5 complexes demonstrate only 21.4% of uptake after 24 h.

### 2.11. Confocal Laser Scanning Microscopy (CLSM)

Uptake studies using confocal scanning laser microscopy (CLSM) were conducted to visualize the uptake of FluoNTC/cNLC complexes. Since the quantitative data show that the concentration of 100 nM was preferred for uptake in 3T3-L1 cells (Figure 11), and also, this was the concentration that showed the highest reduction in lipid droplet production in both ratios (Figure 9 and Figure 10), we used it for microscopy studies. From the obtained images we can conclude that the complexes are indeed taken up by the cells, and they demonstrate vesicular distribution inside the cells (Figure 12A).

After 4 h of incubation, the difference in the uptake between the two ratios is very clear, which is in accordance with our quantitative data. 24 h post-transfection, this difference decreased, as we can see by the number of red-colored vesicles present on both 1:2.5 and 1:5 images. However, the differences in cellular uptake between complexes with different mass ratios are minimal, as they only amount to approximately 5% after 24 h. Furthermore, the obtained z-stacks confirm that the FluoNTC/cNLC complexes are embedded in the cytoskeleton, and are not only adhering to the cellular surface (Figure 12B). To exclude any confounding results due to background fluorescence, we also measured cells that did not undergo any treatment and were used as a control (Figure 12C). No red signal was observed in the untreated cells, which supports the idea that the red signal comes exclusively from the fluorescence group (Cy3) of miRNA.

## 3. Discussion

Recent research has revealed the importance of miRNA in adipocyte development, metabolic function, proliferation, and differentiation [43], and a connection between miRNA expression profile in white adipose tissue and different metabolic parameters, such as BMI, adipogenesis, glycemia, and leptinemia. Many miRNAs regulate glucose and lipid metabolism in physiological and pathological conditions, particularly in adipocyte differentiation, β-cell mass control, and insulin signaling pathways. They affect the regulation of energy balance and metabolic homeostasis [2]. Although using miRNAs as therapeutic systems seems promising and represent an encouraging technological advance, miRNA itself is highly hydrophilic, and its delivery faces multiple extracellular and intracellular barriers [41]. Therefore, developing a proper DDS as a carrier for miRNA is necessary for their delivery.

In this work, we developed cationic NLCs as DDS for miRNA-27a. The composition of cargo-free NLCs was chosen based on the GRAS status (Generally Recognized As Safe) of the used substances and previous good experience regarding the formation of small-sized particles with excellent physical stability [44]. Aiming to improve the stability of NLCs, prevent the aggregation of particles, and consider their lower toxicity levels compared to ionic ones [45], the non-ionic surfactants Tween 80 and Pluronic F68 were added into the formulations. However, as miRNAs are negatively-charged molecules, the production of an NLC formulation with an overall positive surface charge was necessary, which was achieved by adding cationic lipid (OA). A similar approach was evaluated for delivering single-strand oligonucleotides [46,47] and plasmids [48,49,50].

As the strong and rapid binding of miRNAs to cationic carriers is one of the prerequisites for their successful delivery, this study aimed to optimize the formulation of cNLCs and determine the effect of an increasing concentration of OA on the physicochemical properties of produced cNLCs.

Cationic NLCs were produced by the hot HPH process, one of the most beneficial technologies for manufacturing lipid nanoparticles, due to its high effectiveness in reducing particle size and polydispersity, accessible scale-up possibilities, and long-standing use in the pharmaceutical industry [51]. The production conditions were optimized in a previous study [44]. We have succeeded in developing cNLC formulations with increasing concentrations of OA (0–0.5, *w*/*w*), with particle sizes in the range of 110–115 nm and narrow size distribution, keeping in mind that the diameters of NLCs should be closely controlled during preparation so that they are in the range of 100–120 nm [26] (Table 1). Additionally, as ZP plays a crucial role in the complexation of cationic NLCs with miRNA, values between +30 to +50 mV are generally considered suitable for complexation, as they allow electrostatic interactions between the positively charged NLCs and the negatively charged miRNA. Therefore, due to the low ZP values, the cNLC–1 formulation was excluded from further study, while other formulations were in the desirable ZP range (Table 1).

The developed DDS should have several properties; they should protect the miRNA from endonucleases in the body, enable appropriate interaction with the cellular membrane, promote the adequate cellular uptake, release, and distribution of miRNAs, and possess minimal systemic toxicity [16,42]. Therefore, MTS and LDH assays were performed to assess the cytotoxic profile of cationic NLCs, compared to bNLC. Results (Figure 1) have shown that at OA concentrations ≥0.25% (*w*/*w*), dose-dependent cytotoxicity is more pronounced compared to other formulations. Due to the low cytotoxicity profiles, verified additionally with an LDH assay, the cNLC–4, cNLC–5, and cNLC–6 formulations were excluded from further studies (Figure 1). As substances used in cNLC formulations (Precirol ATO^®^ 5, MCT, Tween^®^ 80, and Pluronic F68) have GRAS status and are biocompatible, the toxicity of cNLCs seems to be dependent on the concentration of the used cationic lipid. Previous studies have reported that the toxicity of single-tailed cationic lipids-based delivery systems is related to the migration of OA molecules causing particle interferences and direct interaction with the cellular membrane, which is negatively charged [25].

Long-term stability is critical in pharmaceutical formulations with market potential [52]. For this study, the control bNLC formulation, as well as selected cationic NLC formulations (cNLC–2 and cNLC–3), were stored in transparent glass vials at 5 ± 3 °C, 25 ± 2 °C (60 ± 5% RH), and 40 ± 2 °C (75 ± 5% RH). The formulations were characterized in terms of particle size, PdI, ZP, and pH value, and the results are presented in Figure 2.

Although it was reported that cationic particles have significantly lower stability than negatively charged lipid nanoparticles [53], the cationic NLC formulation cNLC–3 maintained its physicochemical properties even after nine months of storage, regardless of storage conditions. The OA-containing NLCs had better stability than the reported stability of OA-based nanoemulsions, where the particle size was maintained for just one week for all formulations stored at 5 ± 3 °C, and phase separation of systems was noticed after two weeks. These systems were formulated using MCT, Tween 80, and Span 80 [54]. On the other hand, Dukovski et al. reported the good physical stability of OA-containing nanoemulsions composed of Miglyol^®^ 812 and Cremophor^®^ EL over 150 days of storage at 4 °C [55]. In the case of cationic SLNs, the longest reported stability of these systems, composed of Compritol 888 ATO, Pluronic F68, and one cationic lipid, was 180 days [53,56]. In the study of Doktorovova et al., one cSLN formulation consisted of Compritol 888, Poloxamer 188, and CTAB maintained its stability during three months of storage [57]. Although OA-containing NLCs with similar characteristics have been found in the literature (particle size of 125.1 nm and ZP of ~44 mV), a study of the stability of this formulation was not reported [58].

Based on these results, the cNLC–3 formulation was selected as an optimal nanocarrier for miRNA. This formulation had the lowest concentration of OA (0.15%, *w*/*w*), which caused the highest positive surface charge (>40 mV), had optimal particle size (100–120 nm), and was proven to have excellent physical stability over nine months. Given that cell viability exceeding 70% is thought to have non-toxic effects on cells, this formulation, when diluted from 1:250 to 1:50, had no adverse effects on 3T3-L1 cells, according to MTS data [59]. Additionally, the LDH assay showed that the membrane integrity was not affected in all dilutions (<30%) (Figure 1B).

The freshly produced cNLC–3 formulation was used to further prepare miRNA/cNLC complexes. It is known that the miRNA/cNLC ratio affects the cellular uptake and transfection efficiency of miRNA by influencing the complex particle size and net surface charge, hence its stability in biological fluids and ability to associate with negatively charged cell membranes [60]. As a result, to form the miRNA/cNLC complex, the mass ratio between miRNA-27a and CNLC–3 had to be optimized so that the miRNA remained free in solution at the lowest possible concentration. The initial study optimized the incubation time of the complex to 5 min at room temperature. The results (Figure 4) show that in miRNA/OA mass ratios above 1:1 (*w*/*w*), the obtained particle sizes and PdI values were similar compared to those obtained after the production of the cargo-free cNLC–3 formulation. In addition, the PdI of the obtained NLCplexes in these mass ratios was relatively uniform and unaffected by cNLC addition (0.180–0.185). This negligible effect of cNLC addition to the miRNA working solution on particle size and size distribution over a mass ratio of 1:1 could be related to the adsorption of miRNA molecules at the cationic surface of NLCs, which has no significant surface volume charge to change the particle size. In addition, the excess of cNLC molecules indicates good stability of positively charged NLCplexes, since the formed complex did not show any aggregation tendency. Similar findings were obtained by adding DNA plasmids [50] or oligonucleotides [46] to the cationic nanoemulsions.

Furthermore, the results of the surface charge of NLCplexes, obtained by the ELS method, are presented in Figure 3B. The trend of the increasing ZP values of these complexes as a function of the miRNA/OA mass ratio indicates a three-zone colloidal stability model of NLCplex, as described elsewhere [61]. It can be noted that above the mass ratio of 1:2.5, the ZP values reached a plateau with no significant variations in particle size, PdI, and ZP, most likely due to excess cNLC particles and overcharging phenomena that occurred. Similar results have been reported in the literature [39,50].

It is desirable to have a slight-to-moderate excess of positively charged carriers to completely neutralize the negatively charged miRNAs, and ensure that unbound material will not lead to enhanced particle agglomeration or unwanted effects during cell culture experiments. In addition, positively charged complexes can easily interact with the negatively charged cell membrane, thereby increasing cellular uptake and initiating the endocytosis phenomena [62]. Therefore, the mass ratios of 1:2.5 and 1:5 are selected for further analysis. The entrapment efficacy of miRNA in these mass ratios was sufficient (74% and 85%, respectively), as presented in Figure 3B, which was further verified with the absence of a miRNA band during the agarose electrophoresis (Figure 4). These results support our hypothesis that NLCplexes are formed spontaneously due to electrostatic interactions between negatively charged phosphate anchors from the miRNA and positively charged OA molecules from cNLC carriers. A further increase in cNLC concentration above the mass ratio of 1:5 is insignificant, and might even result in adverse cellular effects due to increasing concentrations of unbound cNLC particles.

Figure 5 shows no considerable difference between the physicochemical properties of produced NLCplexes, formed with different miRNA types. These results are essential, as in vitro studies included miRNA-FluoNTC for microscopic experiments and miRNA-NTC, which acts as a non-targeting control in differentiation studies. The miRNA-NTC and miRNA FluoNTC have similar physicochemical properties as miRNA-27a in terms of their structure and the number of nucleotides, but they do not have the therapeutical activity. Therefore, they were used in this study to distinguish the effect between the complexes formed between cNLC and miRNA-27a and cNLC and miRNA-NTC. FluoNTC has a fluorescent-labeled group, which was necessary for all experiments that studied cellular uptake. Furthermore, the results suggest that this DDS is a suitable carrier for other miRNAs with similar physicochemical properties, which may broaden the therapeutic applications for other diseases and conditions.

Cryo-TEM and AFM analyses were conducted to gain more information on the morphology and topography of the selected cNLC formulation and the complex with miRNA, along with the particle size analysis performed by the DLS and LD methods. These methods were considered a complementing technique for lipid nanoparticles due to their potential to be used to study particle shape. Cryo-TEM observations of the cNLC–3 formulation revealed weak circular and ellipsoidal structures representing thin platelets in the top view (Figure 6A). Additionally, the dark needle-like formations are particles viewed edge-on, as the side view increases particle thickness, resulting in a darker appearance. The NLC structures on the top view also have slightly darker spots on the circular structures, probably due to the presence of MCT on the surface of the crystalline particles. The observed particle size of cNLC was nearly 120–140 nm, similar to that obtained by the DLS method, with no apparent signs of aggregation. The formed structures are similar to those already published [63,64].

The AFM method further corroborates the results, in which the topographic investigation revealed thin spherical nanometric particles, as shown in Figure 6B. The produced NLCplexes at a mass ratio of 1:2.5 maintained similar shapes and particle sizes after the complexation of cNLC with miRNA molecules. Because of its small dimensions of roughly 2.5 nm in width and 6–7 nm in contour length [65], cryo-TEM or AFM did not show free miRNA (Figure 6C,D). However, a slightly higher electronic density at the interface of NLC particles may indicate the existence of miRNA adsorbed at the NLC interface (Figure 6C). Neither approach demonstrated that particles agglomerated, although it is reported that NLCs can alter their shape and be more flattened after being deposited on the mica substrate, as the water phase evaporates before measurements.

Stability studies (Figure 7) have shown that the miRNA/cNLC complexes maintain their physicochemical properties for only a few days. Therefore, for the in vitro studies in 3T3-L1 cells, complexes with miRNA-27a, NTC, or FluoNTC were freshly prepared.

The cell viability and cytotoxicity of cargo-free cNLC, free miRNAs (27a and NTC), and miRNA-27a/cNLC complexes in mass ratios 1:2.5 and 1:5 were evaluated with MTS and LDH assay. Although free cNLC has shown a dose-depended decrease in cell viability (Figure 8) due to the positive ZP values, the values were still above 70%. Interestingly, when the same amount of cNLC was complexed with miRNA, cell viability was in the range of 90–100%. This phenomenon may be explained by the phosphate groups in miRNA neutralizing some of the positive charges on OA molecules, reducing the electrostatic interactions of OA with cellular membranes, which were more prominent in cNLC formulations. These findings are consistent with prior research on the toxicity of nucleic acid/cationic liposome complexes [5,66,67]. Thus, the obtained results raise new possibilities for using cNLCs as nucleic acid delivery systems and their application in the 3T3-L1 cell line.

The 3T3-L1 cell line enables a transition investigation from undifferentiated fibroblast-like preadipocytes into mature adipocytes. This process can be achieved by adequately stimulating extracellular adipogenic substances, such as synthetic glucocorticoid dexamethasone (glucocorticoid agonist), insulin, and IBMX (cAMP phosphodiesterase inhibitor), in the presence of FBS to promote differentiation into adipocytes. This cocktail of signals promotes the expression of early regulators (C/EBPβ and C/EBPδ), which activates C/EBPα and PPARγ, and further, a high number of genes involved in the binding, uptake, and storage of fatty acids in adipocytes and their maturation [68]. The process leads to the gradual acquisition of morphological and biochemical characteristics of mature adipocytes, from fibroblast-like to spherical shapes, revealing the accumulation of triacylglycerol in lipid droplets that can be used for staining with the lipophilic dye ORO.

As miRNA-27a is a negative regulator of PPARγ, this adipogenic factor cannot trigger endogenous C/EBPα transcription, which should inhibit adipocyte differentiation [2,11]. The degree of lipid droplet formation in mature adipocytes is investigated by staining the cells with ORO dye on the d_6_ of differentiation to investigate the anti-adipogenic response in transfected cells. For that purpose, miRNA/cNLC complexes were prepared in two mass ratios—1:2.5 and 1:5—to contain 50 nM, 100 nM, 150 nM, and 200 nM of miRNA. Besides miRNA-27a, cNLC was complexed in parallel with miRNA-NTC to disseminate the specific miRNA mimic activity and background effects.

Non-transfected cells (cells only) served as a control to evaluate the general impact of the transfection treatment on adipocyte differentiation and cell vitality. For this purpose, one part of the cells received an induction and differentiation medium with all hormonal substances necessary for cell differentiation, while the other part received only a proliferation medium. The staining intensities of lipid droplets that accumulated ORO dye were assessed with light microscopy (Figure 9) and absorbance measurements (Figure 10). These results have shown a clear difference in the reduction in lipid droplet formation caused by naked miRNA and NLCplexes. Compared to complexes with NTC, significantly lower staining intensities were observed with complexes that contain miRNA-27. Furthermore, it was noticed that the mass ratio had an impact on ORO accumulation, as the reduction in absorbance was greater in complexes with a 1:5 mass ratio, compared to 1:2.5, even though the concentration of miRNA was the same. It can be concluded that free cNLCs also contributed to the process of lipid differentiation, which was confirmed when cargo-free cNLC was applied to cells. However, the reduction in lipid droplet formation was still more efficient when cells were transfected with the miRNA-27a/cNLC complex, compared to miRNA-NTC/cNLC or free cNLC. These methods confirmed that DDS based on cNLC could be used for the delivery of miRNA-27a and the reduction in lipid droplet formation in mature adipocytes.

With regards to the cellular uptake of FluoNTC/cNLC complexes, two significant points could be observed (Figure 11). The internalization in 3T3-L1 cells seems to be time- and concentration-driven. This means that the fraction of complexes that are taken up by cells depends on the aforementioned factors; however, this does not always occur in the manner that would be expected. In the case of our complexes, we can see that uptake is inversely correlated to concentration—a concentration of 100 nM seems to be preferred over 200 nM by cells. The reason behind this might be membrane perturbation caused by the positive charges present in higher numbers at higher concentrations of particles, and the lack of sufficient time for the cells to recover after transfection. This phenomenon has already been documented for other positively charged carriers, and it is related to the interaction of the positively charged particles with the negatively charged extracellular matrix, which activates a cascade of intracellular reactions, resulting in increased membrane permeability and cellular uptake [69]. Another reason for this occurrence might be the involvement of a certain receptor in the internalization of these complexes, which allows uptake only at a certain concentration threshold [70].

On the other hand, there is a direct correlation between uptake and time—no matter what the concentration applied to cells is, the longer the incubation is, the higher the taken-up fraction is as well. The second point of interest is the influence of the positively charged lipid on the interactions with and uptake in cells. Since this part of cNLCs is responsible for interacting with the negatively charged cell membrane, we compared two different mass ratios of FluoNTC/cNLC 1:2.5 and 1:5. What we found out was that a 1:2.5 ratio gives the most promising uptake results in 3T3-L1 cells. Having in mind that the number of positive charges available for interaction with the cell membrane is higher in 1:5 complexes compared to 1:2.5, we can speculate that this difference might lead to higher long-term toxicity [71]. Based on the data obtained from the LDH assay (Figure 8B), it is evident that the complex with a miRNA:cNLC ratio of 1:5, at a concentration of 100 nM, exhibits a higher cytotoxic potential compared to the complex with an miRNA:cNLC ratio of 1:2.5 at the same concentration. One possible explanation for this observation could be the presence of higher concentrations of free cNLC in the 1:5 complex, which could contribute to increased cytotoxicity. This might then explain the decreased uptake of these complexes compared to 1:2.5 complexes.

CLSM studies were done to further confirm the uptake of FluoNTC/cNLC complexes in 3T3-L1 cells. Here, we only analyzed the uptake of 100 nM 1:2.5 and 1:5 complexes, after 4 or 24 h of incubation of cells. This concentration was chosen based on the previously described quantitative data on the uptake, as well as data describing the antiadipogenic effects that these complexes demonstrate as delivery systems for miRNA-27a. As demonstrated in Figure 12A,B, the complexes are efficiently taken up by the cells in a timely manner. The difference in the uptake between 1:2.5 and 1:5 complexes is present and visible (indicated by the white dashed lines in Figure 12A, especially for samples imaged at the 4 h time point). With time, this contrast is reduced. The obtained z-stacks confirm the vesicular distribution and show that the complexes are not just adhering to the surface of the cells, but are indeed inserted in the cytoskeleton (Figure 12B). These findings are in accordance with those demonstrated in Figure 11, where we have quantitatively described the uptake. What is interesting is the uptake pattern of these complexes. They show the vesicular distribution in the cells, which might indicate that their uptake involves one of the endocytic mechanisms. Further studies will focus on elucidating the internalization mechanism that these complexes exploit to enter cells.

## 4. Materials and Methods

### 4.1. Materials

For the production of NLC formulations, octadecylamine (OA) and Tween^®^ 80 were purchased from Sigma-Aldrich (Steinheim, Germany), while Pluronic F68/Poloxamer 188 was obtained from BASF (Ludwigshafen, Germany). Miglyol^®^ 812 (medium-chain triglyceride oil, MCT) and Precirol^®^ ATO 5 (glyceryl distearate/palmitostearate) were obtained from IOI Oleo GmbH (Hamburg, Germany) and Gattefossè Deutschland GmbH (Eschbach, Germany). Ultra-purified water, Milli-Q^®^ water (Millipore SAS, Darmstadt, Germany), was used for NLC preparations.

For the preparation of miRNA/cNLC complexes, three different types of nucleic acids were purchased from Dharmacon (GE Healthcare, Vienna, Austria):double-stranded miRNA mimic mmu-miR-27a-3p (miRNA-27a) with the sequence UUC ACA GUG GCU AAG UUC CGC (MW 13,454 g/mol);miRNA mimic negative control (miRNA-NTC) with the sequence UCA CAA CCU CCU AGA AAG AGU AGA (MW 15,384.3 g/mol);miRNA mimic transfection control with fluorescence-labeled Cy3 group (FluoNTC) and the sequence UCA CAA CCU CCU AGA AAG AGU AGA (MW 15,891.5 g/mol).

Nuclease-free water (RNase-free water), obtained from VWR International (Darmstadt, Germany), was used to prepare the standard miRNA/cNLC complexes under aseptic conditions in a laminar flow box (Herasafe KS, Thermo Fisher Scientific, Austria). The RNase AWAY (Sigma-Aldrich) decontamination reagent was used for surface disinfection.

Tris(hydroxymethyl)aminomethane (TRIS 99%+, for biochemistry) (Acros Organics, Morris Plains, NJ, USA) and NaCl ≥ 99.5%, *p*.a. (Carl Roth, Karlsruhe, Germany) were used for the HPLC determination of unbound miRNA.

To perform in vitro cell culture studies, low-glucose (1 g/L) and high-glucose (4.5 g/L) DMEM with phenol red, as well as low-glucose (1 g/L) DMEM without phenol red (Gibco, Life Technologies Corporation, Paisley, UK), were utilized. Phosphate-buffered saline (PBS; pH 7.4), HEPES buffer solution (1 M), L-glutamine (200 nM), and penicillin/streptomycin (10,000 IU/mL) were also purchased from Gibco, while fetal bovine serum (FBS), human insulin solution, dexamethasone, isobutylmethylxanthine (IBMX), and Triton X-100 were obtained from Sigma-Aldrich.

The CellTiter96^®^ AQueous One Solution Cell Proliferation Assay and CytoToxONE™ Homogeneous Membrane Integrity Assay were purchased from Progema (Madison, WI, USA). Dako fluorescence mounting medium was obtained from Agilent Technologies (Santa Clara, CA, USA). To perform the Oil-Red-O (ORO)-staining procedure, the ORO dye (Sigma-Aldrich), glycerol 85% (Herba Chemosan Apotheker-AG, Vienna, Austria), 2-propanol (VWR Chemicals Prolabo, Paris, France), and formaldehyde 37% (Carl Roth, Karlsruhe, Germany) were used.

For confocal scanning laser microscopy studies (CSLM), the Alexa Fluor 488 Phalloidin and Hoechst 33342 were obtained from Thermo Fisher Scientific (Vienna, Austria), and the formalin solution (neutral buffered, 10%) was from Sigma Aldrich. All chemicals were used as received directly without further purification.

### 4.2. Methods

#### 4.2.1. High-Pressure Homogenization

NLC dispersions were prepared by the hot high-pressure homogenization process, as described elsewhere [72]. Briefly, a mixture of solid lipid (Precirol^®^ ATO 5), liquid lipid (Miglyol^®^ 812), and an increasing concentration of cationic lipid, OA (0–0.5%, *w*/*w*), was melted at a temperature of 5–10 °C above the melting point of the solid lipid until a uniform and clear lipid phase was obtained. Then, a hot aqueous surfactant solution composed of Tween^®^ 80 and Pluronic^®^ F68 was added to the lipid phase and dispersed with a high-shear mixer (Ultra-Turrax^®^ T25, Janke & Kunkel, IKA1-Labortechnik, IKA1-Werke GmbH & Co. KG, Staufen, Germany) at 8000 rpm for one minute. The obtained pre-emulsions were homogenized under the pressure of 800 bar and three homogenization cycles at 70 °C using a *piston-gap* homogenizer (Panda 2K, NS1001L Spezial, GEA Niro Soavi, Parma, Italy). The homogenizer was equipped with a water bath Julabo HE v.2 (Julabo Labortechnik GmbH, Seelbach, Germany) for temperature control. Subsequently, the lipid dispersions were filled in glass vials and cooled in an ice bath for 30 min. The composition of prepared formulations is presented in Table 3.

The NLC formulation without OA (bNLC) was prepared as a control using identical experimental conditions to those described above. In all formulations, the total lipid phase content was constant (5%, *w*/*w*).

#### 4.2.2. Preparation of miRNA/cNLC Complexes

Stock solutions of three types of nucleic acids (miRNA-27a, NTC, or FluoNTC) were prepared in RNase-free water and stored at −80 °C. Working solutions of 1.3 μM nucleic acids were obtained by diluting the stock solutions with RNase-free water, as described previously [42]. The working solutions of miRNA and the selected cNLC formulation were intermixed using equal volumes and vortexed gently (5 s) to obtain standard miRNA/cNLC complexes. These complexes, containing a nucleic acid concentration of 650 nM, were incubated at room temperature for 5 min. The standard miRNA/cNLC complexes were diluted with serum-free, phenol-red-free low-glucose DMEM to obtain concentrations suitable for in vitro transfection studies. The stated in vitro concentrations are always referred to as nucleic acids (miRNA-27a, NTC, or FluoNTC).

To evaluate the miRNA binding capacity and obtain the optimal complexation ratio, a working solution of the miRNA-27a was mixed with the selected cNLC formulation in the following miRNA/OA ratios: 10:1, 5:1, 2.5:1, 1:1, 1:2.5, 1:5, and 1:10 (*w*/*w*). All samples contained 100 nM of miRNA.

#### 4.2.3. Particle Size and Zeta Potential Analysis

Photon correlation spectroscopy (PCS). The hydrodynamic diameter of the mean particle size (z-average), the size distribution of NLC formulations, and the intensity-based particle size of obtained miRNA/cNLC complexes were determined using a Zetasizer Nano ZS (Malvern Instruments, UK). Before measurements in disposable polystyrene cuvettes (PS, semi-micro; Brand GmbH+Co. KG, Germany), NLC formulations were diluted 1:100 (*v*/*v*) with Milli-Q^®^ water to avoid multi-scattering phenomena. The particle size analysis of complexes (100 nM) was conducted in UV micro cuvettes (UVette^®^ 220–1600 nm, Eppendorf, Germany).

Laser diffraction (LD). To detect agglomerates in NLC formulations, a Malvern Mastersizer 2000 with a Hydro 2000 µP sample dispersion unit (Malvern Instruments, Malvern, UK) was used. The results of the particle diameters are presented as volumetric diameters: d_(0.5)_, d_(0.9)_, and d_(0.99)_. The volumetric diameter values represent the percentages of particles possessing a diameter equal to or lower than the given value.

Electrophoretic light scattering (ELS). The zeta potentials (ZPs) of NLC dispersions and miRNA/cNLC complexes were determined using a Zetasizer Nano ZS in a disposable folded capillary cuvette type DTS1070 (Malvern Instruments, Malvern, UK) at 25 °C with a field strength of 20 V/cm. The conversion into the ZP was performed using the Helmholtz–Smoluchowski equation. The measurements of NLCs were performed in distilled water with adjusted conductivity to 50 μS/cm and pH in the range of 5.5–6.0 to avoid fluctuations in the ZP. For the ZP measurements of miRNA/cNLC complexes, the standard complexes were diluted in RNase-free water to reach a final miRNA concentration of 100 nM.

#### 4.2.4. pH Measurements

The pH of the NLC formulations was measured with a pH-meter SevenExcellence (Mettler Toledo, Greifensee, Switzerland) at room temperature.

#### 4.2.5. Stability Studies

The physical stability of the selected NLC formulations was investigated during the storage of formulations at 5 ± 3 °C, 25 ± 2 °C (60 ± 5% RH), and 40 ± 2 °C (75 ± 5% RH) in a climate chamber, model ICH110 (Memmert GmbH + Co.KG, Schwabach, Germany), according to ICH guidelines [73]. The stability studies of NLCs monitored eventual changes in the particle size, PdI, pH, and ZP for nine months.

Physical stability studies of the miRNA-27a/cNLC complexes were performed to monitor potential changes in the physicochemical properties of the miRNA/cNLC complexes. Therefore, complexes were prepared in two different miRNA/OA mass ratios (5:1 and 1:5) and incubated for five minutes at room temperature using a Thermomixer comfort (Eppendorf Austria GmbH, Vienna, Austria). The stability tests of miRNA/cNLC complexes included monitoring particle size, PdI, and ZP using a Zetasizer Nano ZS. For one month, samples were kept in Eppendorf tubes in the fridge at 5 ± 3 °C. The measurements were performed immediately after complex preparation and after 5 min, 1 h, 2 h, and 3 h, as well as after the 1st, 3rd, 5th, 10th, 20th, and 30th days.

#### 4.2.6. Entrapment Efficiency (EE%)

The miRNA loading efficiency in miRNA/cNLC complexes was determined indirectly by reversed-phase high-performance liquid chromatography (RP-HPLC; Agilent 1260 Infinity, Agilent Technologies, Santa Clara, CA, USA). First, using ultracentrifugation (Beckman Coulter Optima XPN-80 Ultracentrifuge, IN, USA), operating at 4 °C and 200,000 × *g* for two hours, the unbound miRNA was separated from the miRNA/cNLC complex. The concentration of unbound miRNA in the supernatant was analyzed with a DIONEX DNAPac PA200, 4.6 × 250 mm column (4 µm) (Thermo Fisher Scientific, Austria). Mobile phase A was 40 mM TRIS (pH 7), whereas mobile phase B was 40 mM TRIS with 1 M NaCl. A gradient was used with a 1.0 mL/min flow rate at 41 °C and an injection volume of 20 µL. The measurement started with 40% B for 1 min and increased to 46.2% B within 8 min, followed by 80% B for 9 min. To detect the miRNA, a wavelength of 260 nm was used. The determined limit of detection (LoD) was 4.0 µg/mL. A standard curve with stock solutions of miRNA (0.5–50 µg/mL) was prepared previously. The percentage of bound miRNA in the complex was calculated according to the following Equation (1):(1)$$EE\left(\%\right)=\frac{W_{\mathrm{total}\mathrm{miRNA}}-W_{\mathrm{free}\mathrm{miRNA}}}{W_{\mathrm{total}\mathrm{miRNA}}}\times 100$$
where W*_free miRNA_* is the amount of unbound miRNA in miRNA/cNLC complexes after ultracentrifugation, and W*_total miRNA_* is the amount of total miRNA in a complex.

#### 4.2.7. Agarose Gel Electrophoresis

The binding ability of miRNA to cNLC was verified by agarose gel electrophoresis. This assay evaluates the eventual migration of unbound miRNA from the complex using the E-Gel Power Snap Electrophoresis System, which is connected to an E-Gel™ Power Snap Camera (Thermo Fisher Scientific Inc., Vienna, Austria). Briefly, miRNA/cNLC complexes in different mass ratios (final miRNA concentration 0.1 µg/µL) were loaded on 4% E-Gel™ agarose gel (Thermo Fisher Scientific Inc., Austria) and run for 15 min, according to the manufacturer’s guidelines. Naked miRNA was used as a control.

#### 4.2.8. Morphology and Topography Studies

Atomic Force Microscopy (AFM). The AFM method was utilized for the examination of the topography of the NLC formulation, as well as miRNA/cNLC complexes. Analysis was conducted on a FlexAFM 5 atomic force microscope (Nanosurf AG, Switzerland) equipped with a C3000 controller. For AFM measurements, cNLC formulation was diluted at 1:1000 in Milli-Q^®^ water, whereas the prepared miRNA–cNLC complexes were incubated for 5 min, and 10 µL of samples were dropped to a freshly cleaved mica substrate. The samples were dried overnight under laminar flow conditions [74]. Height images were recorded in the air at room temperature, and images were obtained in phase contrast mode (tapping mode) with a Tap300Al-G cantilever (Budgetsenzors, Sofia, Bulgaria). All data were processed by Gwyddion software (version 2.55).

Cryogenic transmission electron microscopy (Cryo-TEM). The cryo-TEM method was used to examine the morphology of the cNLC formulation and miRNA–cNLC complex using an FEI Tecnai T-12 microscope (FEI Company, Hillsboro, Oregon, USA) operating at 120 kV. The cryo-TEM preparation of samples was carried out with a Leica^®^ EM GP plunge freezer (Wetzlar, Germany), which allowed for temperature control between 4 and 60 °C and a relative humidity of 99%. The prepared cNLC formulation and standard miRNA/cNLC complexes were diluted 1:1 (*v*/*v*) with RNAse-free water. Then, 3 µL of the samples was vitrified on a TEM support grid (holey carbon film on the copper grid). After removing the excess liquid using blotting paper, the samples were frozen by immersion in cold liquid ethane at around −184 °C to prevent the formation of ice crystals [4]. Data were processed with the Digital Micrograph software (Gatan, Pleasanton, CA, USA) after being captured digitally by Gatan BioScan CCD, Model 792 (Gatan, Pleasanton, CA, USA).

#### 4.2.9. Cell Viability and Cytotoxicity Studies (MTS and LDH Assay)

The cell viability and cytotoxicity of NLC formulations and miRNA/cNLC complexes were evaluated by CellTiter96^®^AQueous One Solution Cell Proliferation Assay (MTS assay) and CytoToxONE™ Homogeneous Membrane Integrity Assay (LDH assay) in mouse embryonic fibroblast-derived 3T3-L1 preadipocytes, according to the manufacturer’s guidelines. For that purpose, cells were seeded on 96-well plates (Greiner Bio-One GmbH, Frickenhausen, Germany) at a seeding density of ~7 × 10^3^ cells/well and incubated the day before transfection to allow the cells to adhere to the wells. After incubation, the medium was removed, and the cells were treated with different dilutions of freshly prepared NLC formulations and miRNA–cNLC complexes (25–100 nM), and then incubated for 4 h.

Then, 25 μL of cell supernatant was transferred to a white 96-well plate (Greiner Bio-One GmbH, Frickenhausen, Germany) and 25 μL of CytoTox Reagent was added to determine the LDH leakage from cells. The plate was incubated for 20–30 min at 37 °C, and 13 μL of stop solution was added to each well. The fluorescence signal was measured at an excitation wavelength of 560 nm and emission wavelength of 590 nm with a UV-VIS plate reader CLARIOstar^®^ Plus (B.M.G. Labtech GmbH, Ortenberg, Germany). Cell cytotoxicity was expressed as a percentage compared to the maximum LDH release in the presence of Triton X-100 (9%, *w*/*v*, positive control). Cells incubated in phenol red-free lgDMEM were used as a negative control. Cytotoxicity was calculated according to Equation (2):(2)$$\mathrm{Cytotoxicity}\left(\%\right)=\frac{F_{\mathrm{test}\mathrm{sample}}-F_{\mathrm{negative}\mathrm{control}}}{F_{\mathrm{positive}\mathrm{control}}-F_{\mathrm{negative}\mathrm{control}}}\times 100$$
where F is the measured fluorescence intensity.

The residual sample volume was removed for the MTS assay, and cells were washed with PBS. Then, 100 µL of fresh serum-free, phenol red-free lgDMEM, and 20 µL (0.2 mg/mL) of the MTS solution were added to each well. The plate was incubated for another 4 h, after which absorbance was measured at 490 nm with the abovementioned plate reader. Cells incubated in culture medium only were used as a control for 100% cellular viability, while 0% of cellular viability was obtained after cell incubation with Triton X-100 (9%, *w*/*v*). Each sample condition was performed at least in triplicate on 96-well plates.

The cell viability (%) was expressed as the percentage of viable cells compared to untreated cells, and was calculated using the following Equation (3):(3)$$\mathrm{Cell}\mathrm{viability}(\%)=\frac{A_{\mathrm{treated}\mathrm{cells}}}{A_{\mathrm{untreated}\mathrm{cells}}}\times 100$$
where A is the measured absorbance of treated and untreated cells.

#### 4.2.10. In Vitro Transfection and Differentiation Studies

The 3T3-L1 preadipocytes were cultivated at 37 °C under a 5% CO_2_ water-saturated atmosphere in a complete proliferation medium consisting of lgDMEM supplemented with 10% FBS, 1% L-glutamine, 1% HEPES, and 1% penicillin/streptomycin. Cells were seeded at 7 × 10^3^ cells/well seeding density into 96-well plates (Greiner Bio-One GmbH, Frickenhausen, Germany) 24 h before transfection.

The transient transfection of miRNA–cNLC complexes was performed in non-confluent 3T3-L1 preadipocytes. First, the standard miRNA–cNLC complexes (650 nM) were prepared and diluted with serum-free lgDMEM, and samples (100 µL) were added to 3T3-L1 cells and incubated for four hours. After the incubation period, 100 µL of complete transfection medium consisting of antibiotic-free lgDMEM supplemented with 20% FBS, 1% L-glutamine, and 1% HEPES was added to samples to obtain final miRNA concentrations from 25 to 100 nM, as described elsewhere [42].

The transfection medium was completely removed after 24 h of incubation, and cells were washed with PBS. A freshly prepared induction medium consisting of high-glucose DMEM (hgDMEM) supplemented with 10% FBS, 1% L-glutamine, 1% HEPES, 1% penicillin/streptomycin, 1% IBMX (500 µM), 0.1% insulin (10 µg/mL), and 0.1% dexamethasone (1 µM) was added to induce the in vitro differentiation of transfected preadipocytes into mature adipocytes extracellularly. After two (d_2_) and four days (d_4_), the medium was renewed with a differentiation medium composed of hgDMEM supplemented with 10% FBS, 1% L-glutamine, 1% HEPES, 1% penicillin/streptomycin, and 0.05% insulin.

After a total differentiation period of six days (d_6_), mature adipocytes were stained with lipophilic Oil-Red-O (ORO) to enable the visualization of oil droplets in cells. First, the differentiated cells were washed with PBS and fixated in 10% formaldehyde (1:10 in PBS) for at least one hour. After removing the fixative, the cells were washed with 100 µL Milli-Q^®^ water twice and incubated with 100 µL 60% (*v*/*v*) 2-propanol (in Milli-Q^®^ water) for 5 min, followed by the complete air-dry of a 96-well plate. Then, 100 µL of prepared and filtered ORO working solution was added to the cells, and after 10 min of incubation, the cells were repeatedly washed with Milli-Q^®^ water. Finally, the cells were covered with 200 µL of 50% (*v*/*v*) glycerol (in Milli-Q^®^ water) to avoid sample drying.

Concentration-dependent transfection series ranging from 25 nM to 100 nM of final miRNA-27a or NTC concentrations were used to assess the influence of miRNA/cNLC complexes on lipid droplet production during adipocyte differentiation. In addition, the sample conditions included control groups, such as single complex components (cNLC, miRNA-27a, and miRNA-NTC) and non-transfected cells referred to as cells only (c_o_). Regarding in vitro transfection and differentiation, the controls were treated the same as the miRNA/cNLC complexes. Tests were performed at least three times for each sample condition.

For the semiquantitative evaluation of accumulated ORO in lipid droplets, absorbance measurements were performed at an excitation wavelength of 500 nm in matrix scan mode (25 × 25) using a CLARIOstar^®^ Plus plate reader (BMG LABTECH, Ortenberg, Germany). Glycerol 50% (*v*/*v*) was used as a blank in the empty wells. The results of absorbance measurements are presented as blank-corrected mean values ± SD.

The lipophilic ORO dye was used to identify lipid droplets in mature adipocytes and to assess the degree of differentiation by staining lipid droplets in mature adipocytes. Light microscopic images were taken using a Leica DMIL microscope Type 090-135.002 (Leica Microsystems GmbH, Vienna, Austria) equipped with a Canon EOS 70D digital camera (Canon GmbH, Vienna, Austria) to compare the staining intensities of ORO between sample conditions.

##### Preparation of ORO Working Solution

A stock solution of ORO was obtained by dissolving 0.35 g ORO dye in 100 mL 2-propanol. The stock solution was stored in the fridge at 5 ± 3 °C. The freshly prepared working solution of ORO dye was obtained by mixing 6 mL ORO stock solution with 4 mL Milli-Q^®^ water. The working solution of ORO dye was stored at room temperature for 30 min and filtered twice through a 0.45 µm disposable syringe filter.

#### 4.2.11. Cellular Uptake Studies

Having in mind that the fraction of the positively charged lipid in the miRNA/cNLC complexes is responsible for miRNA binding and interactions with the cellular membrane, we studied the influence it has on the uptake of these complexes in 3T3-L1 preadipocytes. Namely, two different mass ratios of miRNAcNLC were used—1:2.5 and 1:5, at a concentration of 100 and 200 nM. The concentrations used always refer to the concentration of miRNA in the complex. Cells were seeded in 96-well plates at a density of 7 × 10^3^ cells/well and were transfected after reaching confluency. Uptake was measured at 4 h and 24 h post-transfection. The cells were washed with PBS to remove the fraction of complexes that had not been internalized and re-supplied with phenol red-free lgDMEM. To reach the part of fluorescent complexes inside the cells, we used 0.1% Triton X-100. Fluorescence intensity was then measured at an ex/em of 555/569 nm using a plate reader (CLARIOstar^®^ plate reader BMG LABTECH, Ortenberg, Germany).

#### 4.2.12. Confocal Laser Scanning Microscopy (cLSM)

3T3-L1 cells were seeded at a density of 7 × 10^4^ in glass-bottom dishes (WillCo Wells BV, Amsterdam, The Netherlands) 24 h before transfection. The standard FluoNTC/cNLC complex in mass ratio 1:2.5 was diluted with phenol red-free, serum-free lgDMEM to reach a final miRNA concentration of 100 nM.

After 24 h, the cells were washed with pre-warmed PBS, transfected with 100 µL of samples, and incubated for 4 and 24 h. Then, cells were fixed with a 3.7% formalin solution for 10 min. After removing the fixative and washing cells with PBS, the cells were permeabilized using 0.1 Triton X-100 in PBS, followed by staining them with Alexa Fluor 488 Phalloidin (for the actin cytoskeleton) and Hoechst 33342 (for the cell nucleus). A Dako fluorescence mounting medium was added to reduce the fading of fluorescence during microscopy. The visualization and characterization of the distribution of fluorescent-labeled NTC in complexes were performed on a confocal laser scanning microscope Leica Stellaris 5 (Leica Microsystems Wetzlar GmbH, Wetzlar, Germany). The obtained images were further analyzed using Fiji software 2.9.0.

#### 4.2.13. Statistical Analysis

In the present paper, all data are expressed as the mean ± SD. The significance of differences was evaluated using the ANOVA and Tukey multiple comparisons test at a probability level of 0.05. Differences between means were considered statistically significant for *p* values < 0.05 (*) and highly significant for *p* < 0.001 (***).

## 5. Conclusions

The present work shows that high-pressure homogenization can be used for the successful production of cationic NLCs, with desirable physicochemical properties. Based on a cationic surface charge of NLCs, miRNA/cNLC complexes were produced within several minutes through the electrostatic interaction between the cationic lipid and negatively charged miRNA. Due to the low cytotoxic profile of the formed NLCplexes, and the fast cellular uptake, cNLC represents a suitable DDS for delivering miRNA-27a in 3T3-L1 cells. This complex successfully induced the anti-adipogenic effect and reduced lipid droplet formation in mature adipocytes. Given that this DDS can be used to prepare NLCplexes with different types of miRNAs that have similar physicochemical properties as miRNA-27a, this approach might provide new therapeutic strategies not only to prevent or treat obesity and obesity-related disorders, but also many other diseases and states.

## Figures and Tables

**Figure 1 pharmaceuticals-16-01007-f001:**
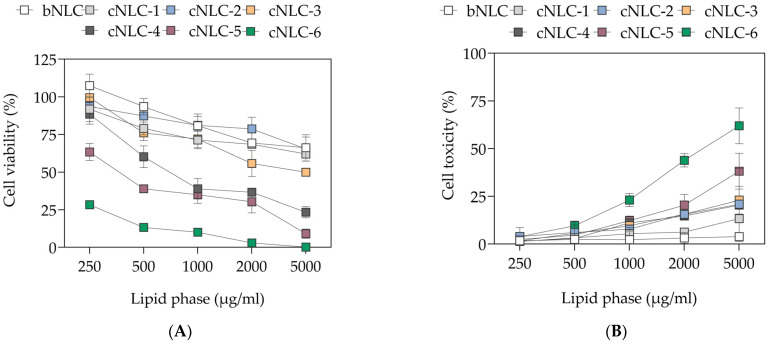
Cytotoxicity on the 3T3-L1 cell line expressed as (**A**) cell viability (%) and (**B**) cell toxicity (%) of NLC formulations containing increasing concentrations of OA (0–0.5%, *w*/*w*). Data are presented as mean ± SD (*n* = 6).

**Figure 2 pharmaceuticals-16-01007-f002:**
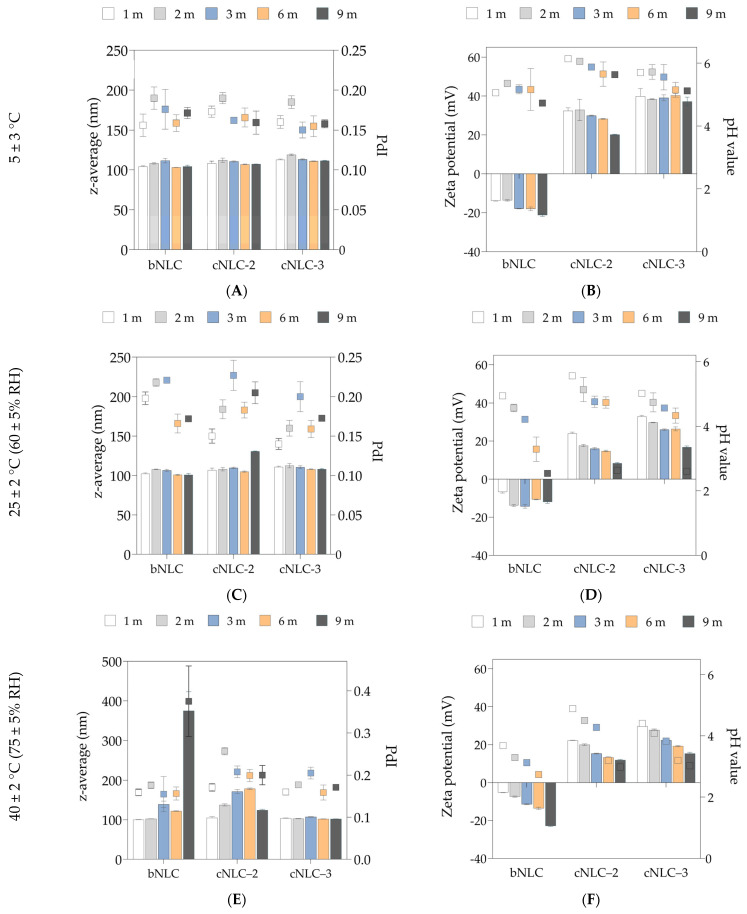
Stability assessment of nanostructured lipid carriers. Measurement of mean size (nm) and PdI values (**A**,**C**,**E**) and zeta potential (mV) and pH values (**B**,**D**,**F**) of NLC after 9 months of storage. Z-average and zeta potential values are presented as bars, while PdI and pH values are given as squares on graphs. Data are presented as mean ± SD (*n* = 3).

**Figure 3 pharmaceuticals-16-01007-f003:**
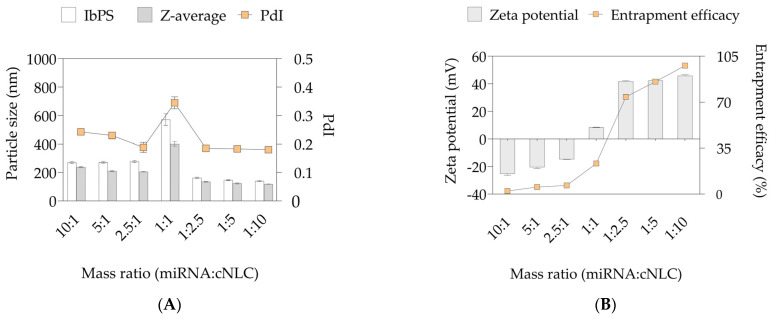
Physicochemical properties of NLCplexes formed in different mass ratios are expressed as (**A**) particle size and PdI and (**B**) zeta potential and entrapment efficiency. Particle size is presented as the mean particle size (orange bars) and the intensity-based particle size (beige bars). NLCplexes contain higher amounts of miRNA-27a (10:1, 5:1, and 2.5:1), equal amounts of miRNA-27a and cationic NLC (1:1), and higher amounts of cationic NLC formulation (1:2.5; 1:5, and 1:10). Data are presented as mean ± SD (*n* = 3).

**Figure 4 pharmaceuticals-16-01007-f004:**
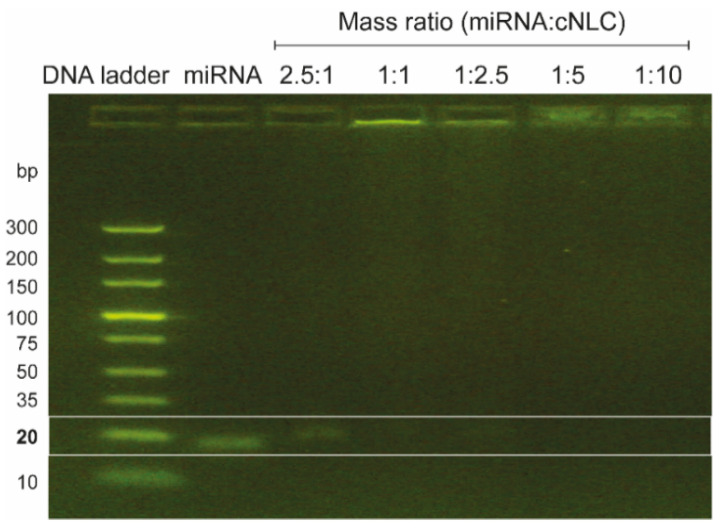
Agarose gel electrophoresis of free miRNA-27a and NLCplexes in different mass ratios. Lane 1: DNA ladder; lane 2: free miRNA-27a; lane 3–7: NLCplexes in miRNA-27a/cNLC mass ratios of 2.5:1; 1:1; 1:2.5; 1:5, and 1:10, respectively.

**Figure 5 pharmaceuticals-16-01007-f005:**
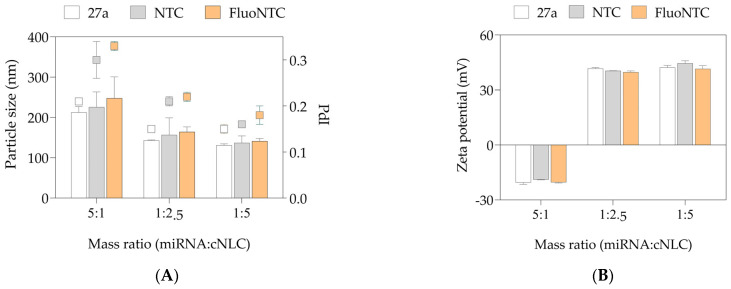
Physicochemical properties of NLCplexes composed of cNLC–3 formulation and different nucleic acids (miRNA-27a, NTC, and FluoNTC), expressed as (**A**) particle size (bars) and PdI (squares) and (**B**) zeta potential values. Data are presented as mean ± SD (*n* = 3).

**Figure 6 pharmaceuticals-16-01007-f006:**
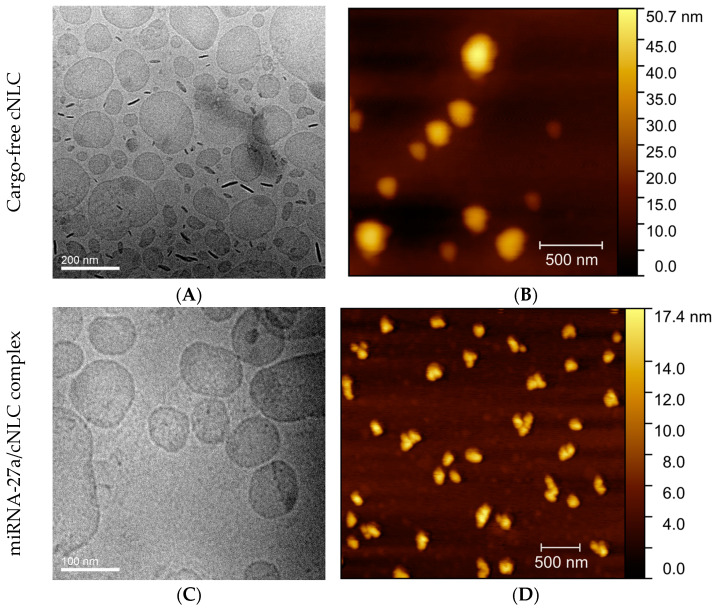
(**A**,**C**) Cryo-TEM images and (**B**,**D**) AFM images of cNLC–3 formulation before (**A**,**B**) and after (**C**,**D**) complexation with miRNA-27a and formation of NLCplexes in mass ratio 1:2.5, respectively. Each panel has its scale bar.

**Figure 7 pharmaceuticals-16-01007-f007:**
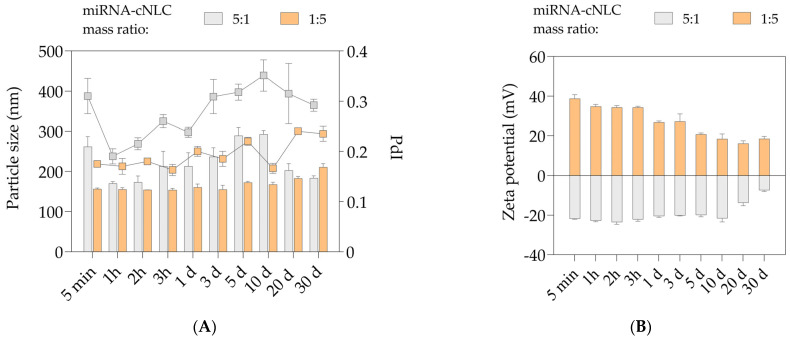
(**A**) Particle size and PdI, and (**B**) zeta potential of miRNA-27a/cNLC complexes over 30 days, stored in the fridge. The complexes were produced in mass ratios 5:1 (beige bars) or mass ratio 1:5 (orange bars). Data are presented as mean ± SD (*n* = 3).

**Figure 8 pharmaceuticals-16-01007-f008:**
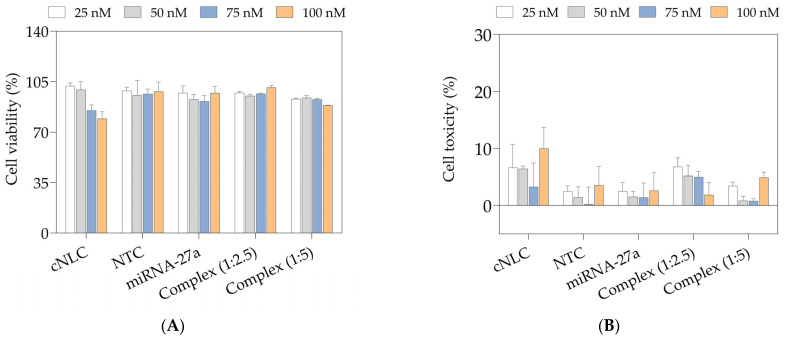
Cell viability (**A**) and cell cytotoxicity (**B**) of cNLC–3 formulation, free miRNA-NTC, and miRNA-27a, as well as miRNA-27a/cNLC complexes in mass ratios 1:2.5 and 1:5 (*w*/*w*). miRNA concentration ranges from 25 nM to 100 nM. Data are presented as mean ± SD (*n* = 3).

**Figure 9 pharmaceuticals-16-01007-f009:**
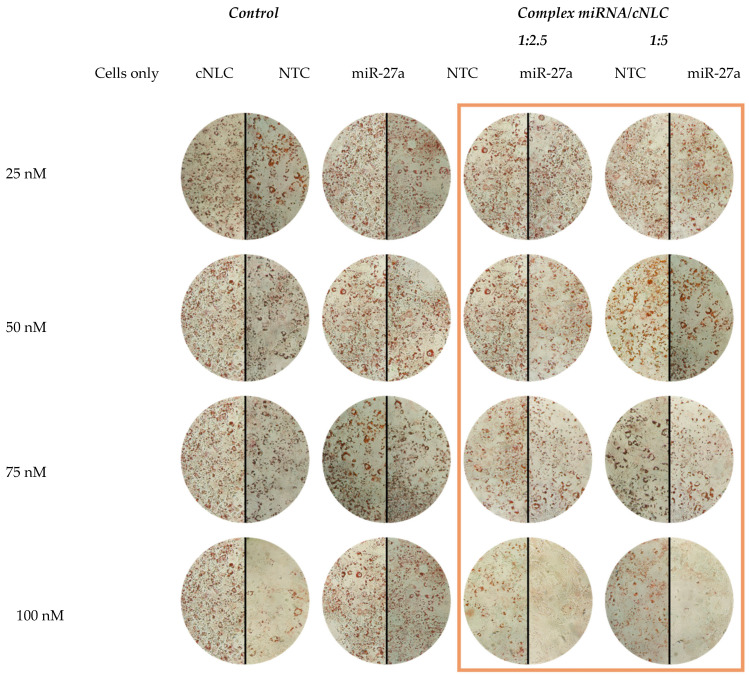
Light microscopic images of transfected and differentiated 3T3-L1 cells on the d_6_ after ORO-staining. Different sample conditions include differentiated cells (cells only), the cargo-free cNLC–3 formulation, free miRNA-NTC, and miRNA-27a in the control group, and miRNA/cNLC complexes containing NTC or miRNA-27a in mass ratios 1:2.5 and 1:5 in the experimental group. Cells only serve as a control to evaluate the effect of the transfection treatment on adipocyte differentiation.

**Figure 10 pharmaceuticals-16-01007-f010:**
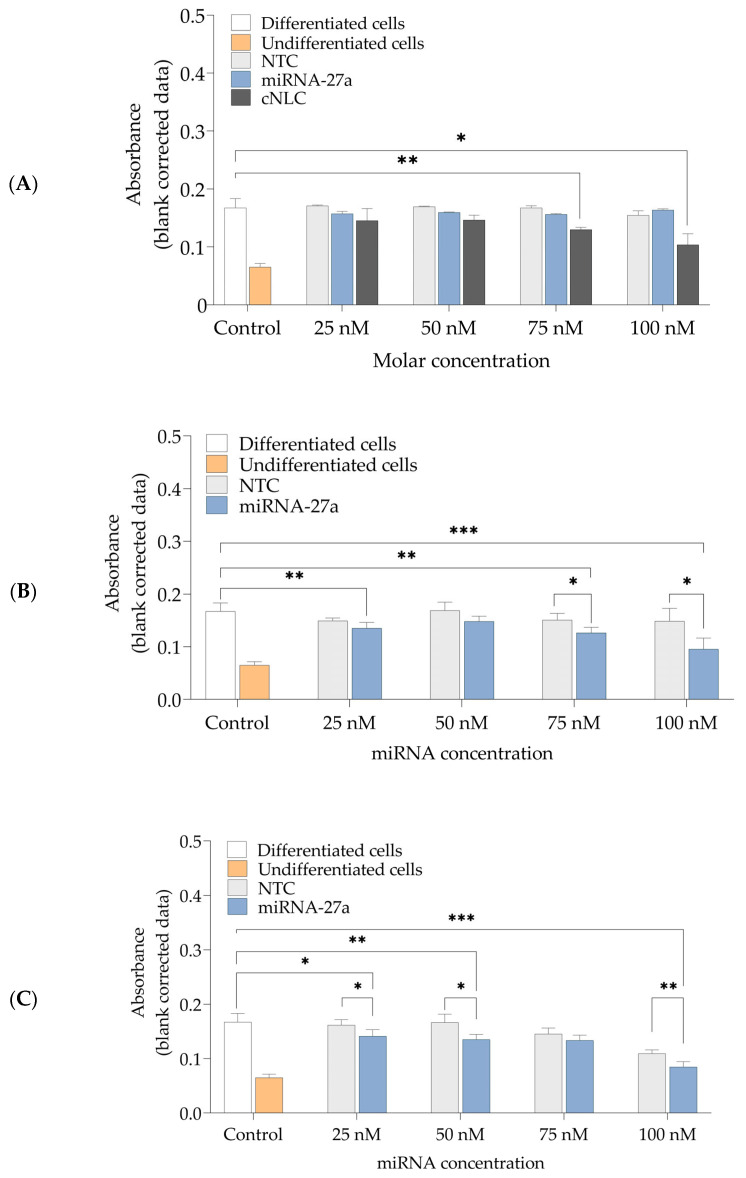
Semiquantitative absorbance measurements of accumulated ORO dye in lipid droplets at d_6_ of differentiation of 3T3-L1 cells. The degree of lipid accumulation is evaluated according to (**A**) controls consisting of differentiated and undifferentiated cells, free NTC, miRNA-27a, and cNLC, and (**B**) miRNA (27a or NTC)/cNLC complexes in mass ratio 1:2.5, and (**C**) miRNA (27a or NTC)/cNLC complexes in mass ratio 1:5. Differentiated and undifferentiated cells serve as a control to assess the impact of the transfection treatment on adipocyte differentiation. The obtained results are presented as mean ± SD (*n* = 6). Statistical significance is given as *p* < 0.05 (*), *p* < 0.01 (**), and *p* < 0.001 (***).

**Figure 11 pharmaceuticals-16-01007-f011:**
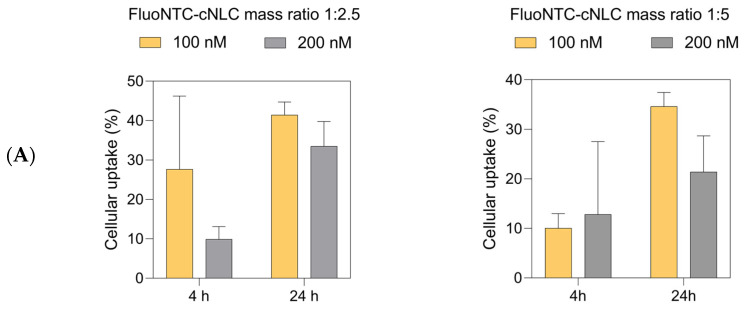
Comparison of intracellular uptake on FluoNTC/cNLC complexes in 3T3-L1 cells. The influence of concentration (**A**) and the influence of fraction of positively charged lipid (**B**) on uptake were evaluated. The obtained results are presented as mean ± SD (*n* = 3).

**Figure 12 pharmaceuticals-16-01007-f012:**
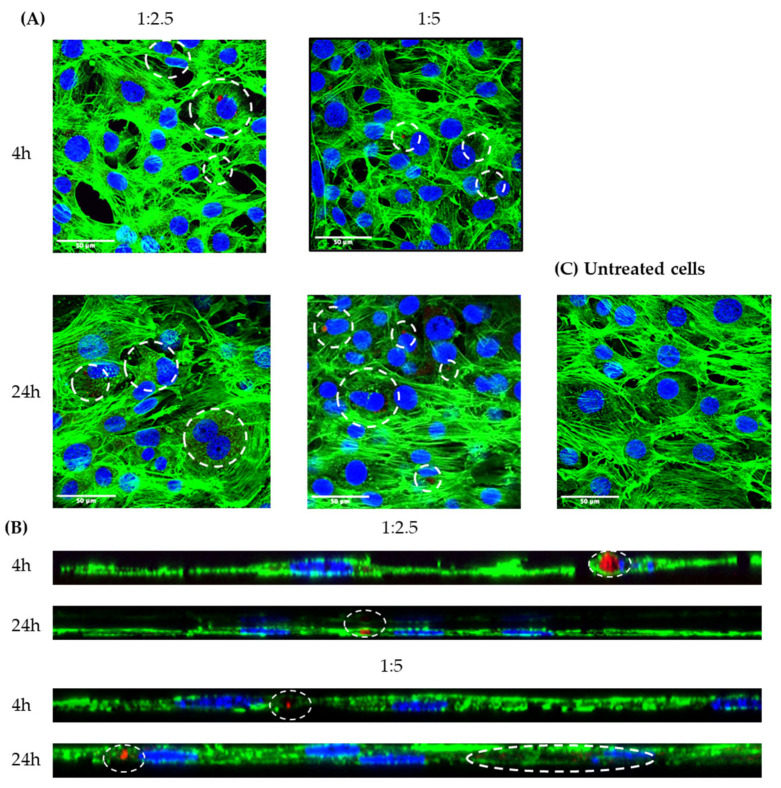
(**A**) Comparison of intracellular uptake of FluoNTC/cNLC complexes in 3T3-L1 cells, using confocal laser scanning microscopy (CLSM). The FluoNTC/cNLC complexes contain Cy3 fluorescence dye (red), while the nucleus is stained using Hoechst 33342 (blue), and the cytoskeleton is counterstained using Alexa Fluor 488 Phalloidin (green). (**B**) The orthogonal projections (z-stacks) of the obtained images. Two different ratios of FluoNTC/cNLC complexes are compared (1:2.5 and 1:5) at two different time points (4 h and 24 h). The concentration used is 100 nM. The internalized complexes are marked on the pictures using white dashed lines. Treated cells are compared to untreated control (**C**).

**Table 1 pharmaceuticals-16-01007-t001:** The PCS data (z-average and PdI), LD data (diameters d_(0.5),_ d_(0.9)_, and d_(0.99)_)_,_ ELS data (ZP), and pH values of NLC formulations with different concentrations of OA (0–0.5%, *w*/*w*) at the day of production (d_0_). Data are presented as mean ± SD (*n* = 3).

Formulation	PCS Data		LD Data (µm)			ELS Data	pH
	z-ave (nm)	PdI	d_(0.5)_	d_(0.9)_	d_(0.99)_	ZP (mV)	
bNLC	105.6 ± 1.8	0.180 ± 0.003	0.128 ± 0.001	0.189 ± 0.002	0.240 ± 0.001	−18.2 ± 1.2	4.998 ± 0.039
cNLC–1	112.9 ± 1.7	0.185 ± 0.009	0.127 ± 0.001	0.188 ± 0.001	0.250 ± 0.002	14.6 ± 0.5	6.706 ± 0.047
cNLC–2	109.4 ± 1.4	0.213 ± 0.015	0.123 ± 0.001	0.187 ± 0.002	0.250 ± 0.001	33.9 ± 1.0	8.390 ± 0.081
cNLC–3	113.2 ± 0.8	0.213 ± 0.007	0.126 ± 0.004	0.184 ± 0.003	0.240 ± 0.013	39.6 ± 2.1	8.861 ± 0.053
cNLC–4	114.9 ± 1.7	0.207 ± 0.006	0.129 ± 0.001	0.188 ± 0.002	0.240 ± 0.003	40.6 ± 0.4	9.114 ± 0.021
cNLC–5	116.4 ± 2.8	0.217 ± 0.016	0.126 ± 0.002	0.186 ± 0.002	0.250 ± 0.004	43.0 ± 2.5	9.323 ± 0.001
cNLC–6	116.2 ± 3.7	0.218 ± 0.013	0.132 ± 0.002	0.183 ± 0.001	0.250 ± 0.009	46.8 ± 2.1	9.617 ± 0.014

**Table 2 pharmaceuticals-16-01007-t002:** Percentage of reduction in ORO dye accumulation in 3T3-L1 cells at d_6_ of differentiation. Data are presented as mean ± SD (*n* = 3).

Percentage of Reduction of Lipid Accumulation in Cells		miRNA/cNLC Complex				Control			
		1:2.5		1:5		NTC	27a	cNLC	Undif. Cells
		NTC	27a	NTC	27a				
miRNA concentration	25 nM	10.77 ± 1.31%	19.16 ± 1.56%	3.11 ± 1.13%	15.45 ± 1.36%	0.77 ± 0.14%	5.98 ± 0.42%	12.87 ± 2.07%	61.08 ± 3.75%
	50 nM	1.01 ± 0.81%	11.49 ± 1.52%	1.53 ± 1.71%	19.04 ± 0.86%	1.17 ± 0.07%	4.49 ± 0.07%	12.39 ± 0.81%	
	75 nM	9.71 ± 0.58%	20.20 ± 1.03%	12.87 ± 1.21%	24.37 ± 1.03%	1.0 ± 0.35%	6.58 ± 0.14%	22.35± 0.51%	
	100 nM	11.08 ± 3.67%	42.87 ± 2.11%	34.71 ± 1.38%	49.40 ± 0.97%	7.48 ± 0.78%	2.09 ± 0.21%	37.9 ± 1.87%	

**Table 3 pharmaceuticals-16-01007-t003:** Composition of produced NLC formulations containing increased concentrations of OA.

Formulation	Lipid Phase (%, *w*/*w*)			Water Phase (%, *w*/*w*)		
	OA	Precirol^®^ ATO 5	Miglyol^®^ 812	Tween^®^ 80	Pluronic^®^ F68	MQ Water
bNLC	0	4.50	0.50	1.33	0.67	93
cNLC–1	0.05	4.455	0.495	1.33	0.67	93
cNLC–2	0.10	4.41	0.490	1.33	0.67	93
cNLC–3	0.15	4.365	0.485	1.33	0.67	93
cNLC–4	0.20	4.32	0.480	1.33	0.67	93
cNLC–5	0.25	4.275	0.475	1.33	0.67	93
cNLC–6	0.50	4.05	0.450	1.33	0.67	93

## Data Availability

Not applicable.

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
