# Peer review of "Development and Characterization of Cationic Nanostructured Lipid Carriers as Drug Delivery Systems for miRNA-27a"

_pharmaceuticals, 2023, doi:10.3390/ph16071007_

Round 1

Reviewer 1 Report

In this study, a miRNA-complexed nanoparticle design was developed as a delivery system for its efficient transfection. Cationic nanostructured lipid carriers (cNLCs) were utilized as the delivery system and all the procedures to prepare the system and its characterization were explained in detail in the experimental part. These complexes were aimed to be used for reducing lipid droplet accumulation in mature adipocytes for better treatment of obesities. 

The following points would help to improve the manuscript for publication: 

-Figure B could not be seen.

-Addition of cationic lipids into the formulation was followed by zeta potential measurements. However, it is still not clear if miRNA molecules were encapsulated inside of the complex, or if some of them adhered on the surface of the complex-structured nano-aggregates, which might affect the release and hence the activity of miRNA molecules tob e delivered in a controlled way. In Figure 4, as feed miRNA content increases compared to cationic lipids, complex miRNA amounts are seen to increase as well. 

-In Figure 6, AFM images with smaller scales could show better if nano-complexes are not making any aggregates. 

In Figure 8B, it would have been explained better how miRNA and complex itself have increased cytotoxicity at 100 nm, while the miRNA-Lpid complex has decreased. 

In this study, a miRNA-complexed nanoparticle design was developed as a delivery system for its efficient transfection. Cationic nanostructured lipid carriers (cNLCs) were utilized as the delivery system and all the procedures to prepare the system and its characterization were explained in detail in the experimental part. These complexes were aimed to be used for reducing lipid droplet accumulation in mature adipocytes for better treatment of obesities. 

The following points would help to improve the manuscript for publication: 

-Figure B could not be seen.

-Addition of cationic lipids into the formulation was followed by zeta potential measurements. However, it is still not clear if miRNA molecules were encapsulated inside of the complex, or if some of them adhered on the surface of the complex-structured nano-aggregates, which might affect the release and hence the activity of miRNA molecules tob e delivered in a controlled way. In Figure 4, as feed miRNA content increases compared to cationic lipids, complex miRNA amounts are seen to increase as well. 

-In Figure 6, AFM images with smaller scales could show better if nano-complexes are not making any aggregates. 

In Figure 8B, it would have been explained better how miRNA and complex itself have increased cytotoxicity at 100 nm, while the miRNA-Lpid complex has decreased. 

Author Response

see attachment 1

Reviewer 2 Report

This manuscript evaluated the cationic lipid (OA)-containing nanocarriers, cNLC, for their delivery of miRNA-27a for potential treatment of adipocyte development. the cNLCs formulations were optimized based on their physical property, stability and cellular toxicity. In addition, further optimizations were also conducted by changing ratio between miRNA and cNLC. lipid droplet formation in mature adipocytes was assessed by staining the cells with ORO dye to investigate an anti-adipogenic effect of miRNA-27a in transfected cells. overall this is an interesting research that target adipocyte for obesity treatment.  However, there are some significant corrections that have to complete.   

1.      the current data did not demonstrate strongly the beneficial effect of cNLC-3/miRNA-27a for preventing adipocyte formation. For example the non-relevant miRNA, i.e. miRNA-NTA, also showed effects of reducing lipid droplet formation at 100 nM (figure9). therefore, more in-depth study is needed to prove the direct correlation between cNLC/miRNA-27a intracellular delivery and inhibition of lipid droplet formation. 

2.      Celluar uptake results(figure 11) showed that higher ratio at 1:5 reduced cell uptaking of cNLC/FluoNTC, compared to those at 1:2.5. author explained the causes were likely due to cNLC toxicity (line 685-687), however, the data of figure8 showed that less or equal cell toxicity between the two ratios. 

3.      Figure 5 A and B were missing

4.      The discussion section seems having a lot repeats of those in the results, therefore, it is better to remove those repeats and cut short considerably .

5.      Did the stability of miRNA-27a have been tested? Although the stability of cNLC/mRNA complex is important as a whole, the release or dissociation of miRNA from cNLC is also critical for the efficacy of inhibition of lipid formations inside cells.

6.      miRNA encapsulation efficiency (EE%) data was missing?

  .   Material and method shall move forward as section 2.

2.      Line144/145, “pH of cNLC was adjusted to ~ 7 to produce theoretically fully ionized OA molecules at the interface of NLC particles”: was pH adjustment prior to addition of miRNA? how pH adjustment influence of EE% or stability? What were pH after miRNA addition? 

3.      Figure1A, .corrections is needed for X-axis title.

4.      In all table and figures, values of n and error bars (S.D or SEM) need to be specified.

5.      Line172/173, “This decrease in the ZP values can be attributed to the coverage of the positively-charged NLC interface with the negatively-charged free fatty acids and interaction with OA molecules”. However, Figure2B/D/F showed that ZPs of bNLC, without positive charged components, was impacted by increase temperatures as well.  what are the causes? 

6.      Line181/182, “As the cNLC−3 formulation has shown an excellent physical stability profile,…”. cNLC-3 demonstrated physical stability at 5C over times, like bNLC and cNLC2 did. The reason of choosing Cnlc-3, but not cNLC2, need to be justified.

7.      Figure 6A, what those dark rod-like substance were?

8.      Line284-286, “ It can be concluded that even though NLC carriers (cNLC−3 formulation) have- shown excellent physical stability for nine months, the complex with miRNA can maintain its physicochemical properties for just a couple of days…” could that instability attribute to RNA dissociated from cNLC-3? what are the possible causes for instability?

9.      Figure 8B, Y scale need to be adjusted to ~15% instead of 30%

1.    In figure9, it seems that miRNA-NTC/cNLC complexes at 100 nM showed a reduced ORO staining signal at 1:2.5 and 1:5. Why was that?

1.     In Figure 10A, it seems that cNLC alone at 75 nM and 100 nM were able to induce ORO absorbance compared to differentiated control cells. While after complexation with miRNA-27a at 1:2.5 and 1:5 ratio, the ORO absorbance in lipid droplet decreased at various RNA concentrations. Was those decrease attributed to cNLC alone or to miRNA functioning? What were the cNLC concentration in B and C?

.    .  Line 379-381 “As shown in Figure 10 (C), the incubation of cells with miRNA 27a/cNLC complexes decreased the absorbance from 0.141 ± 0.03 (25 nM) to 0.085 ± 0.010 380 (100 nM), which led to reduction of lipid formation for 24.37 ± 1.03% (75 nM) and 49.40 ± 381 0.97% (100 nM),”  however, in Figure10C, why miRNA-27a/cNLC at 75 nm did not show lipid droplet reduction?

1.   Figure 11 A/B, for 100 and 200 nm complexation, what were the concentration of cNLC used? Could it be due to the toxicity of cNLC, causing lower cell uptake at 1:5 compared to 1:2.5?. however, CLSM image of Figure 12 seems not supporting the uptaking kinetics behaviors b/w 4hr vs. 24 hr and 1:2.5 and 1:5.

.  .   Line572-580, authors compared their DDS with LNP delivery systems. There was no data to compare the two. 

 L  line584, “… which acts as a non-targeting control in differentiation studies”, what did this mean by non-targeting? was there any targeting function of cNLC? 

Author Response

see attachment 2

Reviewer 3 Report

pharmaceuticals-2443573

Development and characterization of cationic nanostructured lipid carriers as drug delivery systems for miRNA-27a

The manuscript by Tucak-Smajić et al. described the development of cationic nanostructured lipid carriers (cNLCs) for miRNA-27a delivery. The authors demonstrated that the cNLCs reduced lipid droplet accumulation in mature adipocytes, which would be potential in the treatment of obesity and related disorders. The authors may consider the following comments to improve the manuscript.

1. The authors should summarize previous approaches to deliver miRNA-27a and determine the research gap that results in the need to develop a new nanocarrier.

2. Methods: the authors should cite relevant references to support the presented methods in section 4.

3. Particle size: the authors should present only the z-average, which is sufficient for interpretation and discussion.

4. Figure 1: please remove the non-English text. The X-axes should be revised to reflect the correct time scale.

5. Figure 5 is empty.

6. Some graphs are of low quality (Figures 1, 2, 3, 7, 8, and 10). Please improve them.

7. Please include the release data to support statements in lines 577-580.

 Minor editing of English language required

Author Response

see attachment 3

Round 2

Reviewer 3 Report

The manuscript was appropriately revised. However, Figure 10 is still of low quality, probably due to some technical issues, as stated by the authors. Please fix it before publishing.

Author Response

All figures are re-uploaded in the manuscript
